# Highly conductive tissue-like hydrogel interface through template-directed assembly

Jooyeun Chong[1,4], Changhoon Sung [2,4], Kum Seok Nam [2], Taewon Kang[1], Hyunjun Kim [1], Haeseung Lee[1], Hyunchang Park[1], Seongjun Park [2,3] & Jiheong Kang [1,3]

Over the past decade, conductive hydrogels have received great attention as tissue-interfacing electrodes due to their soft and tissue-like mechanical properties. However, a trade-off between robust tissue-like mechanical properties and good electrical properties has prevented the fabrication of a tough, highly conductive hydrogel and limited its use in bioelectronics. Here, we report a synthetic method for the realization of highly conductive and mechanically tough hydrogels with tissue-like modulus. We employed a template-directed assembly method, enabling the arrangement of a disorder-free, highly-conductive nanofibrous conductive network inside a highly stretchable, hydrated network. The resultant hydrogel exhibits ideal electrical and mechanical properties as a tissue-interfacing material. Furthermore, it can provide tough adhesion (800 J/m$^2$) with diverse dynamic wet tissue after chemical activation. This hydrogel enables suture-free and adhesive-free, high-performance hydrogel bioelectronics. We successfully demonstrated ultra-low voltage neuromodulation and high-quality epicardial electrocardiogram (ECG) signal recording based on in vivo animal models. This template-directed assembly method provides a platform for hydrogel interfaces for various bioelectronic applications.

Bioelectronic devices that interface with biological tissues have received considerable attention due to their importance in neuroscience, diagnostics, therapy, and wearable and implantable devices[1–5]. However, the electrodes used in most bioelectronic devices are still rigid and dry, in contrast to soft and water-rich biological tissues. To avoid inflammatory response from applying rigid electrodes, conductive hydrogels are employed since they have soft mechanical properties. However, conductive additives such as metal, CNT, or graphene cannot be employed since they cause adverse reactions in the body due to their cytotoxicity. Therefore, conductive polymer hydrogel is considered a promising material for tissue-interfacing electrodes due to its biocompatibility, tissue-like mechanical properties, mixed electron/ion conduction, and water-rich nature[1,6,7].

Over the last decade, conductive polymer hydrogels have been studied extensively to develop an ideal material for the tissue-interfacing electrode[7–10]. Among them, significant progress has been made in poly(3,4-ethylenedioxaythiophene):poly(styrene sulfonate) (PEDOT:PSS)-based materials due to their relatively high electrical conductivity[11–14]. PEDOT:PSS conductive hydrogels are typically

[1]Department of Materials Science and Engineering, Korea Advanced Institute of Science and Technology (KAIST), Daejeon 34141, Republic of Korea. [2]Department of Bio and Brain Engineering, Korea Advanced Institute of Science and Technology (KAIST), Daejeon 34141, Republic of Korea. [3]KAIST Institute for NanoCentury, Daejeon 34141, Republic of Korea. [4]These authors contributed equally: Jooyeun Chong, Changhoon Sung. e-mail: spark19@kaist.ac.kr; jiheongkang@kaist.ac.kr

fabricated by one of two methods. The first method involves the design of a pure PEDOT:PSS network through additive engineering with ionic liquids[7,15], glycerol, or dimethyl sulfoxide (DMSO) solvent[12]. The resultant pure PEDOT:PSS hydrogel exhibits a relatively high electrical conductivity (~40 S/cm) due to the presence of large aggregation of PEDOT domains. Unfortunately, these aggregates result in undesirable mechanical properties such as a high Young's modulus (~1 MPa) and low stretchability (~20%). In addition, this hydrogel is unable to form robust adhesion with biological tissues, resulting in unstable operation in long-term applications[16–18]. The second method combines a soft and stretchable hydrated polymer with PEDOT:PSS to achieve soft and stretchable mechanical properties[11,19–21]. Although this

hydrogel exhibits high stretchability (> 100%), it lacks electrical conductivity (<1 S/cm) due to the insufficient long-range assembly of the PEDOT network in the presence of the soft insulating polymer[11]; hence, there is a trade-off between mechanical and electrical properties in PEDOT:PSS hydrogels. To the best of our knowledge, no studies have developed an approach for simultaneously realizing soft, stretchable, and tough hydrogels with high electrical conductivity of more than 10 S/cm.

Here, we report a novel template-directed assembly method and conduction mechanism to realize a highly conductive, soft, and tough hydrogel interface with tough tissue adhesion for suture- and adhesive-free bioelectronic applications (Fig. 1a). The key point of this

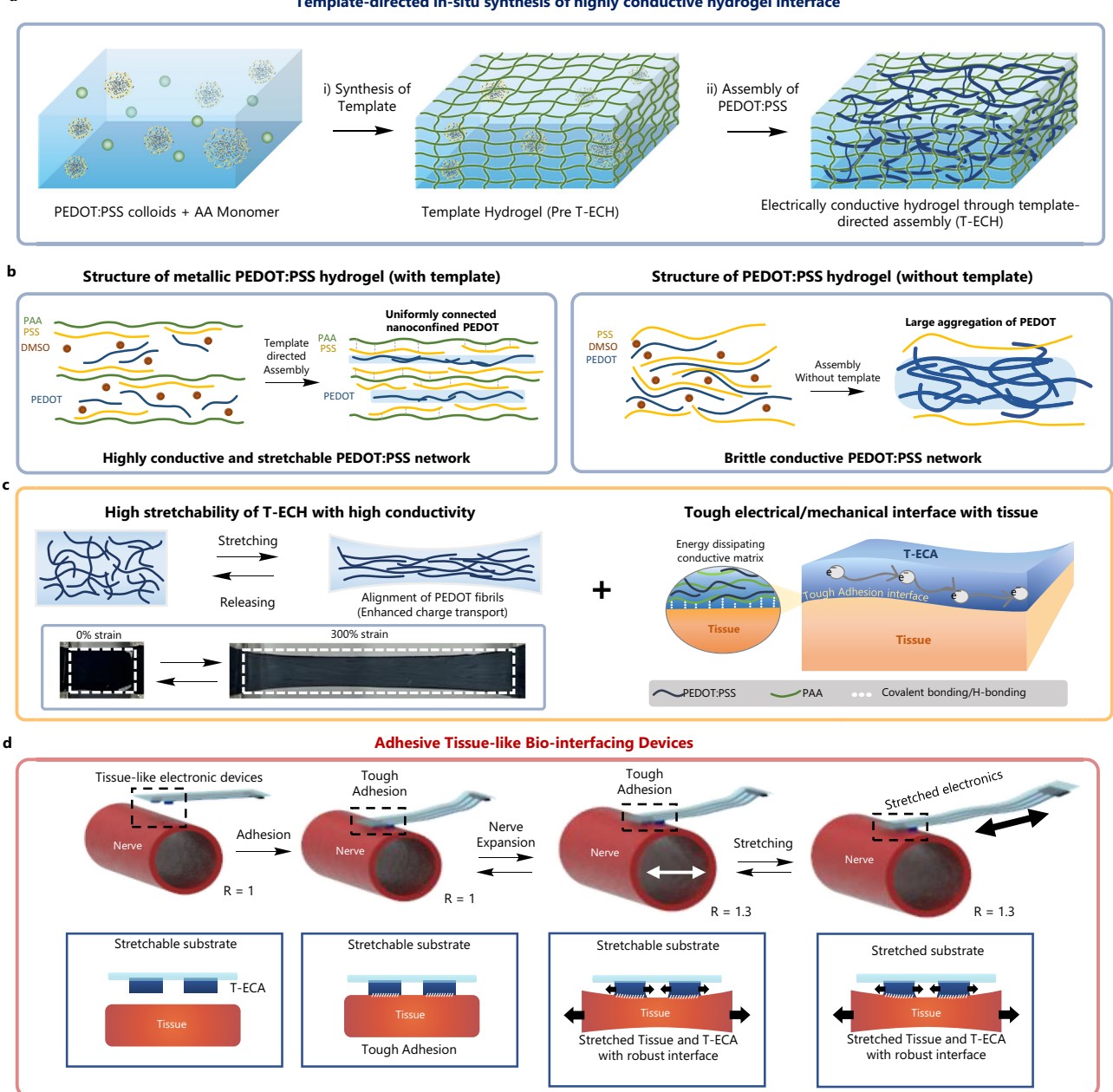

**Fig. 1 | Design and synthesis of T-ECH for bio-interfacing devices. a** Schematic illustration of the template-directed in-situ synthesis of T-ECH. In Pre T-ECH, right after the template polymer (PAA) synthesis, PEDOT:PSS is in the colloidal state. After a drying and annealing process with DMSO solvent, T-ECH is made, having a highly conductive PEDOT network inside the template network. **b** Schematic illustration of the PEDOT:PSS network structure in the hydrogel with (left) and without (right) template polymer. **c** Schematic illustration and optical image of highly stretchable T-ECH (left). When stretched, PEDOT:PSS fibers align to the direction of the applied strain. Schematic illustration of tough adhesion of T-ECH with biological tissue (right). **d** Schematic illustration of tissue adhesive bio-electronic device made of T-ECA (adhesive T-ECH) electrode on the stretchable and dynamic biological tissue.

strategy is the post-assembly of the PEDOT:PSS fibers in the nano-confined space of the chemically crosslinked soft polymeric template network. The polyacrylic acid (PAA) template forms multivalent hydrogen bonds with the PSS shell to enable the growth of an ultrathin PEDOT:PSS fibrous network along the PAA chains with significantly reduced energetic disorders (Fig. 1b). Consequently, our hydrogel, referred to as T-ECH (electrically conductive hydrogel through template-directed assembly), shows a tissue-like modulus (25 kPa), high stretchability (610%) (Fig. 1c), high toughness (1 MJ/m$^3$), high water content (90 wt%) and a record-high conductivity (247 S/cm), which is two orders of magnitude higher than that of previously reported stretchable PEDOT:PSS hydrogels (Supplementary Fig. 1). In addition, T-ECH achieved strong adhesion on wet tissues via hydrogen bonding from PAA, covalent bond formation with tissue, and efficient energy dissipation of the PEDOT:PSS network (Fig. 1d). Using T-ECH, we demonstrate advanced ultralow-voltage neuromodulation and stable bioelectronic signal recording with high signal to noise ratio (Fig. 1e), while retaining tough adhesion and excellent biocompatibility. Due to its excellent mechanical, electrical, and adhesive characteristics, T-ECH is presented as an ideal conductive hydrogel interface material for various bioelectronic applications (Supplementary Fig. 2).

## Results

### Template-directed assembly of highly conductive PEDOT:PSS hydrogels

In general, PEDOT:PSS colloids in an aqueous solution can transform into a conductive fibrous network through dry annealing and anisotropic re-swelling process with various additives and water[12,22,23]. During the dry-annealing process, the PEDOT chains start to form strong π–π interactions and create a conductive assembled network. This process results in a conductive hydrogel with an interconnected fibrous conductive PEDOT network surrounded by anionic PSS shells. Therefore, a relatively high electrical conductivity of ~40 S/cm was achieved due to the large, interconnected PEDOT domain stabilized by the anionic PSS shell[12]. However, bulk aggregates of PEDOT chains are inevitably formed, which results in a high Young's modulus (1 MPa), low stretchability (~30%), and low mechanical toughness. We envisioned that such bulk aggregate structures have high conformational and energetic disorder, which significantly limit charge transport properties.

To avoid the bulk aggregation of PEDOT chains, we utilized the template-directed assembly of a highly conductive nanofibrous PEDOT network. Template-directed assembly is a method that uses a nanoporous membrane as a template to make nanostructures from materials such as conductors, semiconductors, and carbon[24–26]. We hypothesized that a template network with a high binding affinity with the PSS shell could suppress the heavy aggregation of PEDOT chains through a three-dimensional confinement effect and induce a homogeneous nanofibrous conductive network along the template polymer chains (Fig. 1b). Without a template network, PEDOT chains tend to form large domains of bulk aggregates with high conformational and energetic disorder.

T-ECH was synthesized as follows (Fig. 1a): First, an acrylic acid (AA) monomer solution was homogeneously mixed with a PEDOT:PSS solution (PEDOT:PSS solute content is 10 wt% relative to AA). Then, the PAA template network was formed by either photo or thermal radical polymerization of AA monomers with the crosslinker N,N′-methylene bisacrylamide (MBAA). In this state (Pre T-ECH), PEDOT:PSS is in colloidal form with hydrophobic PEDOT in the core surrounded by an anionic PSS shell. Subsequently, DMSO was added to the Pre T-ECH to induce the transformation of PEDOT:PSS from a colloids to extended nanofibers by disturbing the ionic interaction between PEDOT and PSS. Once the linear PEDOT chains are made, they can be connected to each other through π–π interaction. To facilitate their connection, all

the solvents are removed through dry-annealing to make PEDOT fibers encounter each other. Therefore, a uniformly connected PEDOT network is assembled along the PAA template chains. Finally, it was re-swelled in water to form a highly conductive and mechanically tough hydrogel (T-ECH).

### Nanoconfined homogenous electrical network in T-ECH

Pre T-ECH, which has colloidal PEDOT:PSS inside the PAA hydrogel, is too soft to handle, whereas Pure PEDOT:PSS hydrogel (Pure ECH), made only with PEDOT:PSS, is stiff and brittle (Fig. 2a). In contrast, T-ECH, which has a well-connected PEDOT:PSS fibrous network inside the PAA-template network, exhibits a set of exceptional mechanical properties, including tissue-like Young's modulus (25 kPa), high strain at break (610%), high toughness (1 MJ/m$^3$), and increased fracture toughness (Fig. 2b, c and Supplementary Fig. 3). These mechanical characteristics are the result of the double-network structure of T-ECH, which has an energy-dissipating long-range PEDOT:PSS nanofibrous network and a stretchable PAA network[27,28]. These two networks can coexist homogeneously and behave synergistically due to the multivalent hydrogen bonds between PAA and PSS. When other template were used, such as polyacrylamide (PAAm) and poly(2-hydroxyethyl methacrylate) (pHEMA), which do not form strong interactions with PSS, significant phase separation was observed (Supplementary Fig. 4a–c). Due to the low binding affinity of template network and PEDOT:PSS, they showed low fracture strain and serious phase segregation of the template network and PEDOT:PSS.

In addition, similar to other double-network hydrogels, T-ECH can endure high compressive stress (Fig. 2d, e and Supplementary Movie 1). In contrast, due to its low toughness, the single-network PAA hydrogel was completely smashed under a compressive stress. When a large strain is applied to T-ECH, the fibrous PEDOT:PSS network significantly absorbs and dissipates mechanical energy through its network reconfiguration and continuous breakage/reformation of hydrogen bonds with the PAA template. Therefore, this soft, stretchable PAA network can significantly deform and maintain its mechanical structure without fracturing.

In order to use electrode materials in bioelectronics, conductive hydrogels must have sufficiently high electrical conductivity and strain-insensitive electrical properties for stable recording and stimulation in vivo[7,29]. Our T-ECH electrode exhibited a record-high conductivity (247 S/cm) among conductive polymer hydrogels, even in the presence of 90 wt% water content and 9 wt% nonconductive PAA content (Supplementary Fig. 5a, b). As the PEDOT:PSS content was only 1 wt% in the hydrogel, this suggests that the PEDOT:PSS network in T-ECH exhibits superior electrical conduction without conformational and energetic disorders. Therefore, the electrical resistance of T-ECH measured was significantly low (Supplementary Fig. 5c). Moreover, the electrical resistance of T-ECH was insensitive to more than 200% stain and cyclic strain (Fig. 2g and Supplementary Fig. 6). The resistance of T-ECH even decreased slightly when stretched to 100% due to the alignment of PEDOT chains along the direction of strain[12,22]. Not only the electrical properties, but the mechanical properties also had reversible characteristics, having low residual strain during the cyclic tensile test (Supplementary Fig. 7). In addition, the properties of T-ECH can be modified by adjusting the amounts of crosslinker and PEDOT:PSS (Fig. 2c and Supplementary Table 1).

However, it is somewhat counterintuitive that T-ECH exhibits higher electrical conductivity than Pure ECH even with less PEDOT:PSS content. The major reason for its high electrical performance is the elimination of bulk PEDOT aggregates in the percolation network and the formation of a thin, extended fibrous structure of PEDOT chains (Fig. 3a). In Pure ECH, PEDOT and PSS bulk domains are inevitably formed. Thus, it has microscale PEDOT:PSS-rich aggregates, as confirmed by microcomputed tomography (Supplementary Fig. 8). Since Pure ECH has a large number of PSS aggregates that disconnect the

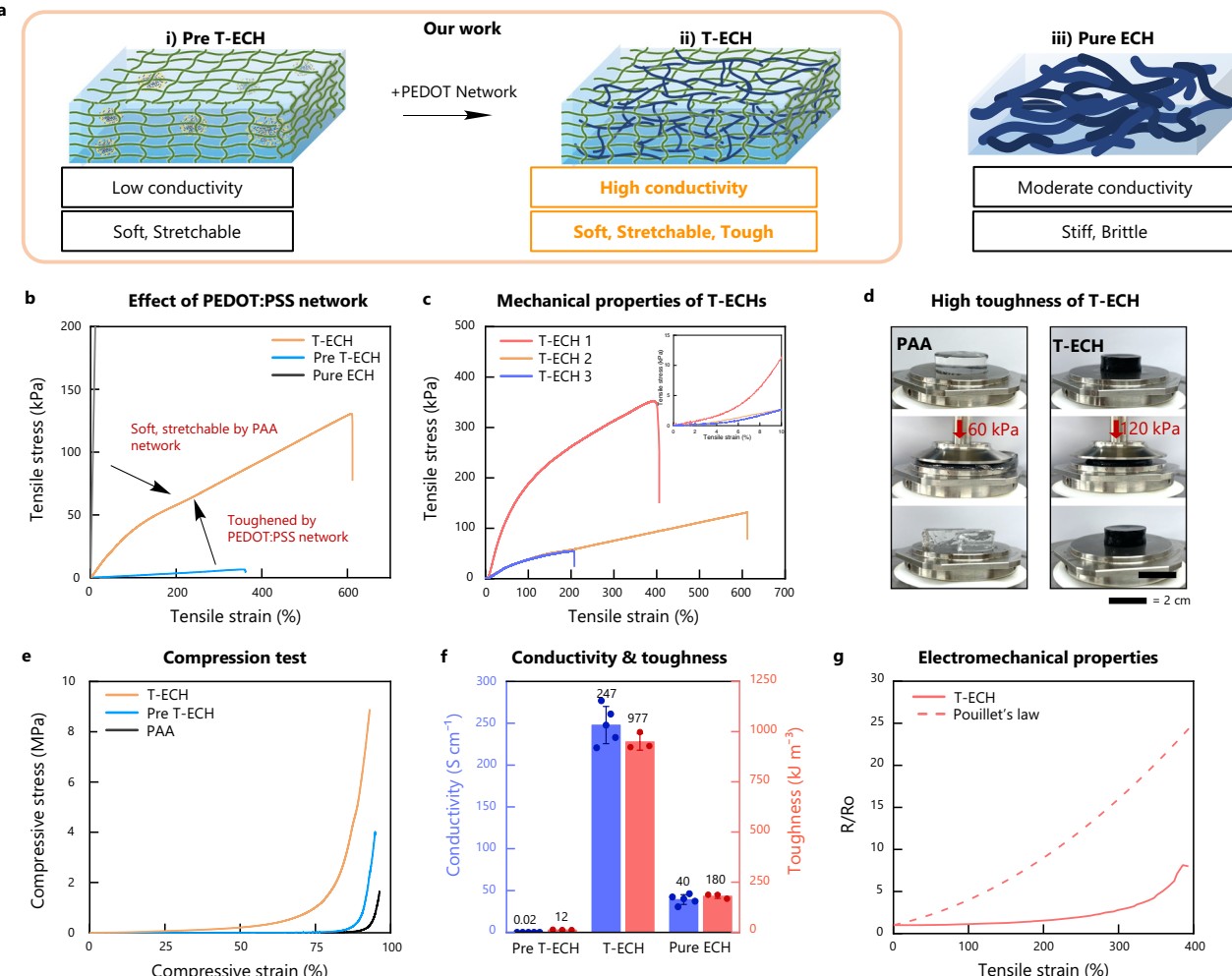

**Fig. 2 | Mechanical and electrical properties of conductive hydrogels.**
**a** Schematic illustration of Pre T-ECH, T-ECH, and Pure ECH. T-ECH has a double network structure of template polymer (PAA) and PEDOT:PSS. Pre T-ECH lacks PEDOT:PSS network while Pure ECH lacks a template network. **b** Stress-strain curves of T-ECH, Pre T-ECH, and Pure T-ECH (Strain rate = 50%/min). Mechanical properties of T-ECH are modified by changing the amount of crosslinker (MBAA) and PEDOT:PSS content (Supplementary Table 1). **c** Stress-strain curves of T-ECH with various PEDOT:PSS contents and crosslinking concentrations. **d** Images of

compressed PAA and T-ECH with 60 kPa (20 N) and 120 kPa (35 N), respectively.
**e** Compressive stress-strain curves of T-ECH, Pre T-ECH, and PAA (Strain rate = 0.2%/min) **f** Graph comparing electrical conductivities (and toughness of Pre T-ECH, T-ECH, and Pure ECH. The data plotted represents the mean and standard deviation ($n = 5$ for conductivity, $n = 3$ for toughness, n means number of independent experiment). **g** Electrical resistance change of T-ECH under strain (Strain rate = 200%/min).

PEDOT domains, the PSS-rich domains significantly block the conductive path. Therefore, the PEDOT-PEDOT interdomain hole transfer is limited. Also, highly entangled PEDOT networks with extensive conformational disorder work as bottlenecks of efficient charge transport in Pure ECH[30]. Consequently, the conductivity cannot exceed 40 S/cm due to the limited transfer path. In sharp contrast, T-ECH shows a homogeneous PEDOT:PSS nanofibrous network without microscale phase separation inside the hydrogel (Supplementary Fig. 8b). During the synthesis of T-ECH, both PEDOT and PSS fibers grow only through the porous PAA template network. Therefore, PAA template physically confines the PEDOT:PSS fibers in T-ECH to enable a well-connected fibrous network. Accordingly, insulating PSS doesn't make large domains that block the conductive PEDOT pathway. Therefore, continuously connected PEDOT path enables high electrical conductivity. In addition, the fine-extended PEDOT:PSS fibers have fast and efficient charge transport via efficient intra-chain charge transport (Supplementary Figs. 9 and 10)[30–32]. The intra-chain charge transport in Pure ECH is inefficient due to the entangled bulk PEDOT chains. However, T-ECH has linear and extended chain conformations leading to fast intra-chain charge transfer.

Here, PAA can induce a linear PEDOT network inside the porous structure of the template polymer through strong hydrogen bonding with PSS. However, other template polymers that lack sufficiently high binding affinities with PSS cannot induce a linear PEDOT network, resulting in low conductivity. For example, T-ECH made of a PAAm template showed low electrical conductivity of 0.01 S/cm due to the macroscopic phase separation of PAAm and PEDOT:PSS domains (Supplementary Fig. 4a, b). Therefore, strong interaction between the PSS shell and the template polymer is required to produce a homogeneous network with high conductivity.

To investigate the structure and morphology, hydrogels were studied using small-angle neutron scattering (SANS) (Fig. 3b, c). We could obtain information about the gel structure using power-law decay, $I - q^{-n}$ (Supplementary Fig. 11 and Supplementary Table 2)[33–35]. At small angles ($q < 0.01 \, \text{Å}^{-1}$), the structure of Pure ECH was found to have a thick fibrous assembly of PEDOT chains (>600 Å), whereas a much thinner fibrous assembly of PEDOT chains was formed in T-ECH. At mid-range angles ($0.03 \, \text{Å}^{-1} < q < 0.1 \, \text{Å}^{-1}$), we found that Pure ECH had the most compact structure, and T-ECH had linearly extended PEDOT chains in the soft PAA network. Moreover, PAAm-based T-ECH was

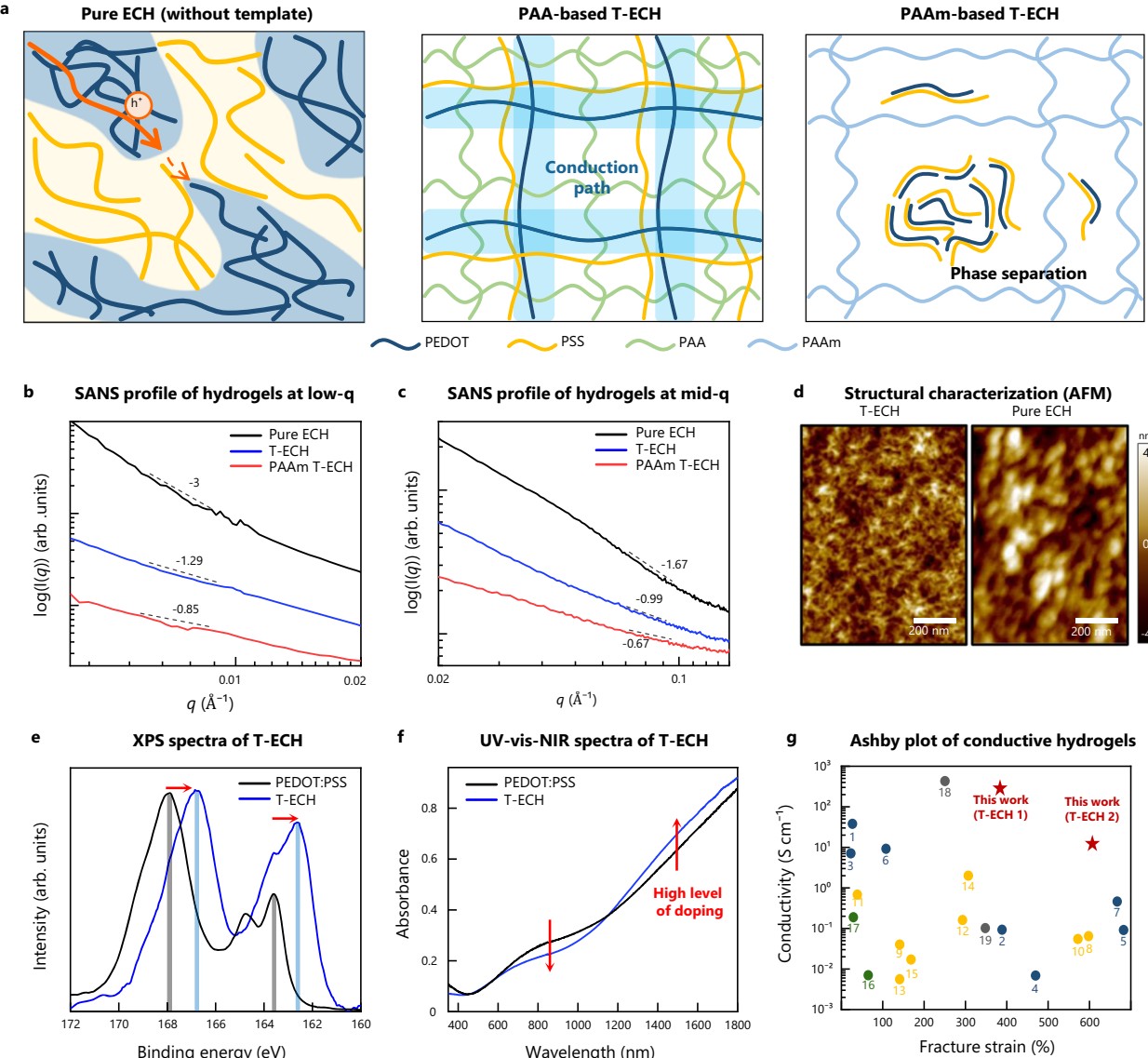

**Fig. 3 | Morphological and structural characterizations of T-ECH. a** Schematic illustration of microstructures of Pure ECH, T-ECH, and PAAm-based T-ECH (PAAm T-ECH). In Pure ECH, the electrically conductive pathway is disrupted by insulating PSS-rich domains. On the other hand, T-ECH has a continuous PEDOT-connected network without bulk PSS aggregates limiting the conductive pathway. PAAm-based T-ECH cannot have PEDOT fibrous network due to the phase separation of PEDOT:PSS and PAAm. **b** SANS measurements of Pure ECH, T-ECH (T-ECH 1), and PAAm-based T-ECH at low-$q$ ranges (<0.02 Å$^{-1}$). **c** SANS measurements of Pure ECH,

T-ECH (T-ECH 1), and PAAm T-ECH at mid-$q$ ranges (0.02 Å$^{-1}$ < $q$ < 0.14 Å$^{-1}$). **d** AFM height images of T-ECH (left) and Pure ECH (right). Similar results were observed in two independent samples. **e** XPS spectra of the fully dried films of PEDOT:PSS (black) and T-ECH (blue). **f** UV-vis-NIR absorbance spectra of PEDOT:PSS (black) and T-ECH (blue). **g** Ashby plot of conductive hydrogels. PEDOT:PSS-based hydrogel (blue), polyaniline (PANI)-based hydrogel (yellow), polypyrrole (PPy)-based hydrogel (green), and metal-based hydrogel (gray) (Supplementary Fig. 1).

found to lack a PEDOT-assembled network. In addition, to obtain conformational change from Pre T-ECH to T-ECH, small-angle X-ray scattering (SAXS) was conducted (Supplementary Fig. 12). It was found that DMSO annealing and re-swelling transformed colloidal PEDOT:PSS to thin fibrous PEDOT:PSS network, making ordered PEDOT:PSS/PAA lamella structure. This conformational change was further seen in atomic force microscopy (AFM) image (Supplementary Fig. 13), confirming that PEDOT:PSS fibers are made in T-ECH.

As the interactions between PEDOT chains are enhanced in the dried film state[36,37], we investigated the π-π distance between PEDOT chains through wide-angle X-ray scattering (WAXS) of fully dried hydrogel films (Supplementary Fig. 14). The Pure ECH peak around $q = 1.85$ Å$^{-1}$ arose from π-π stacking of PEDOT thiophenes, whereas the π-π stacking peak of T-ECH occurred at a higher $q$ (1.87–1.91 Å$^{-1}$), indicating a closer PEDOT-PEDOT distance (3.36–3.28 Å). Thus, the

PEDOT-PEDOT interactions were stronger and more well-ordered in T-ECH, which resulted in a more stable and homogeneous linear PEDOT network with significantly decreased conformational disorders. Due to the increased interchain packing of PEDOT, it could have a high interchain charge transfer. Consequently, T-ECH can have high electrical conductivity through facilitated inter- and intra-chain charge transfer through continuously connected PEDOT chains. Furthermore, T-ECH showed stable electrical conductivity even after five weeks (35 days) in deionized water and phosphate-buffered saline (PBS) (Supplementary Fig. 15).

We further confirmed the uniform, thin network structure of PEDOT:PSS in T-ECH with AFM (Fig. 3d and Supplementary Fig. 16). From the height and amplitude images, we could clearly observe that T-ECH had a thin fibrous structure, whereas the Pure ECH had bulk aggregates. In addition, scanning electron microscopy (SEM) and

transmission electron microscopy (TEM) images agreed with our findings, showing bulk layer-like structures in Pure ECH and a fine, homogeneously interconnected structure in T-ECH (Supplementary Figs. 17 and 18).

Subsequently, we investigated the PEDOT:PSS ratio and its chemical states in T-ECH using X-ray photoelectron spectroscopy (XPS) measurements of the dried hydrogel films (Fig. 3e). The XPS spectra of T-ECH showed clear peaks at ~167 and 162.5 eV due to the S $2p$ spectra of PSS and PEDOT moieties, respectively. Compared with PEDOT:PSS, T-ECH had a drastically increased ratio of PEDOT to PSS from 0.53 to 0.88, implying that an excess amount of insulating PSS shell was washed out during the template-directed assembly of the highly conductive PEDOT:PSS fibrous network[38]. Surprisingly, in T-ECH, the S $2p$ peaks of both PEDOT and PSS exhibited a large shift to lower energies, which has not been reported previously. We assume that the red shift of PEDOT (from 163.5 to 162.5 eV) is due to the high doping level of thin PEDOT chains[39–43]. Because the thin PEDOT chains in T-ECH have a more accessible three-dimensional area than the large PEDOT aggregates in PEDOT:PSS, they interact more with polyanions. Therefore, they can be stabilized more effectively by anionic PSS. In addition, the red shift of the S $2p$ peak of PSS was due to the hydrogen bond interactions between the PAA template and the PSS shell. Moreover, the structural difference of Pre T-ECH and T-ECH was investigated through XPS analysis (Supplementary Fig. 19). It was found that Pre T-ECH had colloidal bulk PEDOT:PSS. Therefore, it was confirmed that the DMSO annealing process made the fibrous PEDOT:PSS network inside PAA.

Finally, we investigated the doping level of PEDOT chains in T-ECH using ultraviolet-visible-near infrared (UV-vis-NIR) spectroscopy (Fig. 3f). There was a decrease in the polar band absorbance (7–800 nm) and an increase in the bipolar band absorbance (above 1100 nm), indicating a higher charge carrier concentration in T-ECH than that in PEDOT:PSS[22,44]. As a result, highly doped nanofibrous PEDOT greatly increased the conductivity of hydrogel. Therefore, to the best of our knowledge, our T-ECH is the only conductive-polymer hydrogel to have both high electrical conductance of over 100 S/cm and stretchability of over 200% strain with tissue-like mechanical properties (Fig. 3g).

## Tissue adhesion of T-ECH

For bidirectional bioelectronic communication, it is crucial to establish conformal and stable electrical contact between the device and target tissue[16]. Our T-ECH electrode was designed to directly form tough adhesion onto wet tissues with high conductivity. Therefore, intimate bidirectional communication between electronic devices and tissues is realized without an additional adhesion layer (Fig. 4a).

When T-ECH contacts wet tissue surfaces, the PAA intimately forms multivalent hydrogen bonding arrays and creates stable physical crosslinks with the tissue surface. In addition, due to the energy-dissipating double-network structure of T-ECH, it exhibited strong interfacial toughness with tissue (400 J/m$^2$) (Fig. 4b). Both Pre T-ECH (no PEDOT network) and PAA hydrogels can also make multivalent hydrogen bonding arrays with tissue; however, because of the absence of the PEDOT:PSS energy dissipation network, they could not realize tough adhesion with tissue (<125 J/m$^2$). Moreover, neither hydrogel is electrically conductive.

To achieve long-term stable adhesion under harsh environments, including ions, water, and temperature, we incorporated a covalent crosslinking strategy based on amide coupling. The carboxylic acid group of PAA in T-ECH cannot undergo an amide coupling reaction with the amino groups of tissue spontaneously because the hydroxy group is a poor leaving group. Therefore, we activated the carboxylic acid groups of PAA using 1-ethyl-3-(3-dimethylaminopropyl)carbodiimide (EDC) and N-hydroxysuccinimide (NHS esters). The resulting activated T-ECH-adhesive (T-ECA) can form covalent crosslinking

bonds with various tissue surfaces. We observed enhanced toughness (800 J/m$^2$) and shear strength (120 kPa) after covalent crosslinking. In addition, there was no change in the electrical properties after the functionalization of T-ECA with T-ECH. Furthermore. T-ECA maintained robust adhesion on porcine skin for more than two weeks in an aqueous environment (Fig. 4c).

To evaluate the adhesion performance of T-ECA with various wet surfaces, we conducted a 180° peeling test for interfacial adhesion energy and a lap shear test for various tissues with polyethylene terephthalate (PET) backing (Fig. 4d and Supplementary Fig. 20a–g). T-ECA formed intimate and tough adhesion to various wet tissues (750 J/m$^2$ for arteries, 250 J/m$^2$ for kidneys) within 10 s of applying gentle pressure (Fig. 4d, e). Moreover, T-ECA showed high adhesion to amine-functionalized gold, suggesting conformal integration of T-ECA with electronic devices (Supplementary Fig. 20h). In summary, T-ECA is a highly conductive adhesive with a high-water content (~90%), tissue-like softness (Young's modulus of 25 kPa), high stretchability (610%), and high toughness (1 MJ/m$^3$).

## Bioelectronic functionalities

We have demonstrated the robust compatibility and functionality of T-ECH electrodes for bioelectronic applications. The desirable mechanical properties of T-ECH enable robust and conformal interfaces with neural tissues. In addition, the strong adhesive ability allows the T-ECA electrodes to adhere to the target tissue without any suturing or an additional non-conductive adhesive layer.

The electrochemical properties of T-ECH were investigated after soaking in PBS (Supplementary Fig. 21). The impedance of the T-ECH electrodes was significantly lower than that of commercial metal electrodes (27.8 Ω at 1 kHz), which was attributed to the high electronic conductivity and large electrochemical surface area of the T-ECH electrodes. In addition, we found that the impedance of the T-ECH electrode were strain-insensitive, up to 100%. Furthermore, the T-ECH electrodes exhibited electrochemical stability for seven days in PBS, with stable impedance and CSC values. The T-ECH electrodes exhibited robust qualities for electrical stimulation. The measured charge storage capacity (CSC) of 80 mC/cm$^2$ was much higher than those of platinum and iridium oxide (IrOx) electrodes (Supplementary Fig. 21). The electrochemical properties of T-ECH demonstrate the potential of T-ECH electrodes for recording and stimulation applications.

The performance of T-ECH electrodes was evaluated using in vivo neuromodulation and electrophysiological recording (Fig. 5a, c). Sciatic nerve stimulation was successfully conducted using T-ECH electrodes. Due to their mechanical properties and strong adhesion with biological tissue, the T-ECH electrodes were able to maintain stable electrical interfacing under continuous leg motion. T-ECH electrodes allow for ultra-low-voltage neuromodulation in vivo (Fig. 5a, b). Electrically evoked leg movements were achieved with ultra low-voltage stimulation (40 mV at 1 Hz) without penetrating the nerve or removing the epineurium (Supplementary Movie 2), which is one of the lowest stimulus voltages reported to the best of our knowledge[7,45]. An increase in electrically evoked leg movements was observed with increasing stimulation voltage, which plateaued at 400 mV.

The T-ECH electrodes also demonstrated robust recording capability (Fig. 5c). Due to the improved adhesive properties of T-ECH electrodes, stable interfacing between the electrodes and the wet, dynamic environment of a continuously beating heart is possible without additional fixatives. In combination with its high conductivity, the T-ECH electrodes were able to achieve stable, high-quality recordings of electrocardiogram (ECG) signals with a signal-to-noise ratio (SNR) of 1009 (60.08 dB) (Fig. 5d). This exceptionally high SNR can be obtained with only bare T-ECHs compared to other ECG devices[46,47].

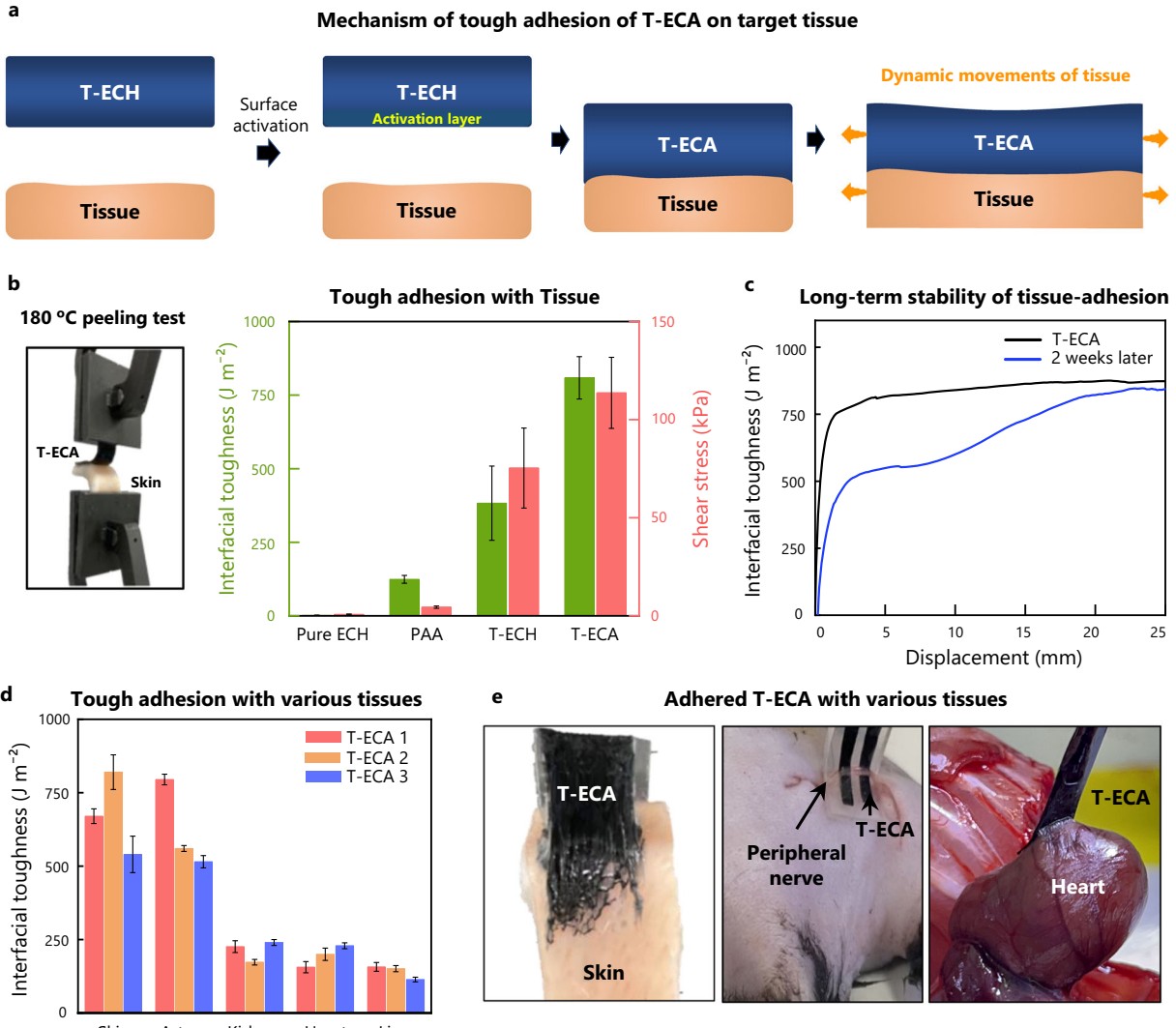

**Fig. 4 | Adhesive properties of T-ECH and T-ECA. a** Schematic illustration of adhesion mechanism of T-ECA. T-ECA is made by activating the adhesive layer on the surface of T-ECH. **b** 180 degrees peeling test, and shear test conducted with pig skin. The data plotted represents the mean and standard deviation ($n = 3$, $n$ means the number of the independent experiment). **c** 180 degrees peel test conducted two weeks after applying on the porcine skin. **d** 180 degrees peeling test of T-ECA hydrogels activated from T-ECH hydrogels, with various wet tissues including porcine skin, artery, kidney, heart, and liver. The data plotted represents the mean and standard deviation ($n = 3$, n means the number of the independent experiment). **e** Optical camera images of strongly attached T-ECA on various tissues.

## Biocompatibility

To assess the potential for long-term bioelectronic interfaces, we evaluate the in vivo biocompatibility of T-ECH-based implants by wrapping bare T-ECH film around the sciatic nerve of mice. To assess the biocompatibility of the T-ECH, two control groups were tested. The same surgical procedure without the implantation of T-ECH electrodes was conducted on a sham group and the second group received rigid stainless steel film implants. After 2 weeks of implantation, the immune response and variation in the cell population of the sciatic nerve tissue were observed by immunohistochemistry (IHC) with the following markers; S-100 for Schwann cells, neurofilament medium (NFM) for neurofilaments (axons), and Iba-1 for macrophages (Fig. 6). The T-ECH implants exhibited no significant difference in the population of Schwann cells and axons compared to the sham group. In contrast, the rigid stainless steel implant group demonstrated a significantly decreased population of myelin-forming Schwann cells due to the compression and occlusion by the pressure of the rigid substrate[48] (Fig. 6a, c, d). Moreover, there was no significant difference in the expression of Iba-1, a key indicator of the immune response, between the T-ECH and sham groups (Fig. 6b, e), whereas rigid stainless steel implants evoked an increased immune response compared with the sham and T-ECH groups in consistency with previous reports[6,49,50]. Similarly, the highest degree of fibrosis was observed in the rigid stainless steel group with hematoxylin and eosin (H&E) staining (Supplementary Fig. 23).

## Discussion

In this paper, we have made a highly conductive and mechanically tough percolation pathway in soft hydrogel through a template-directed assembly method. The high binding affinity between the template network and insulating PSS is crucial for the formation of a highly conductive PEDOT network without conformational and energetic disorders. Consequently, T-ECH, our hydrogel, exhibits the highest electrical conductivity with high stretchability among the previously reported conductive polymer-based hydrogels. Also, the double network structure in T-ECH enables soft, stretchable, and tough mechanical characteristics. We could further activate T-ECH to T-ECA by applying EDC and NHS at the interface to obtain high interfacial toughness with diverse wet tissues. T-ECH also showed excellent in-vivo biocompatibility because of its tissue-like softness and high

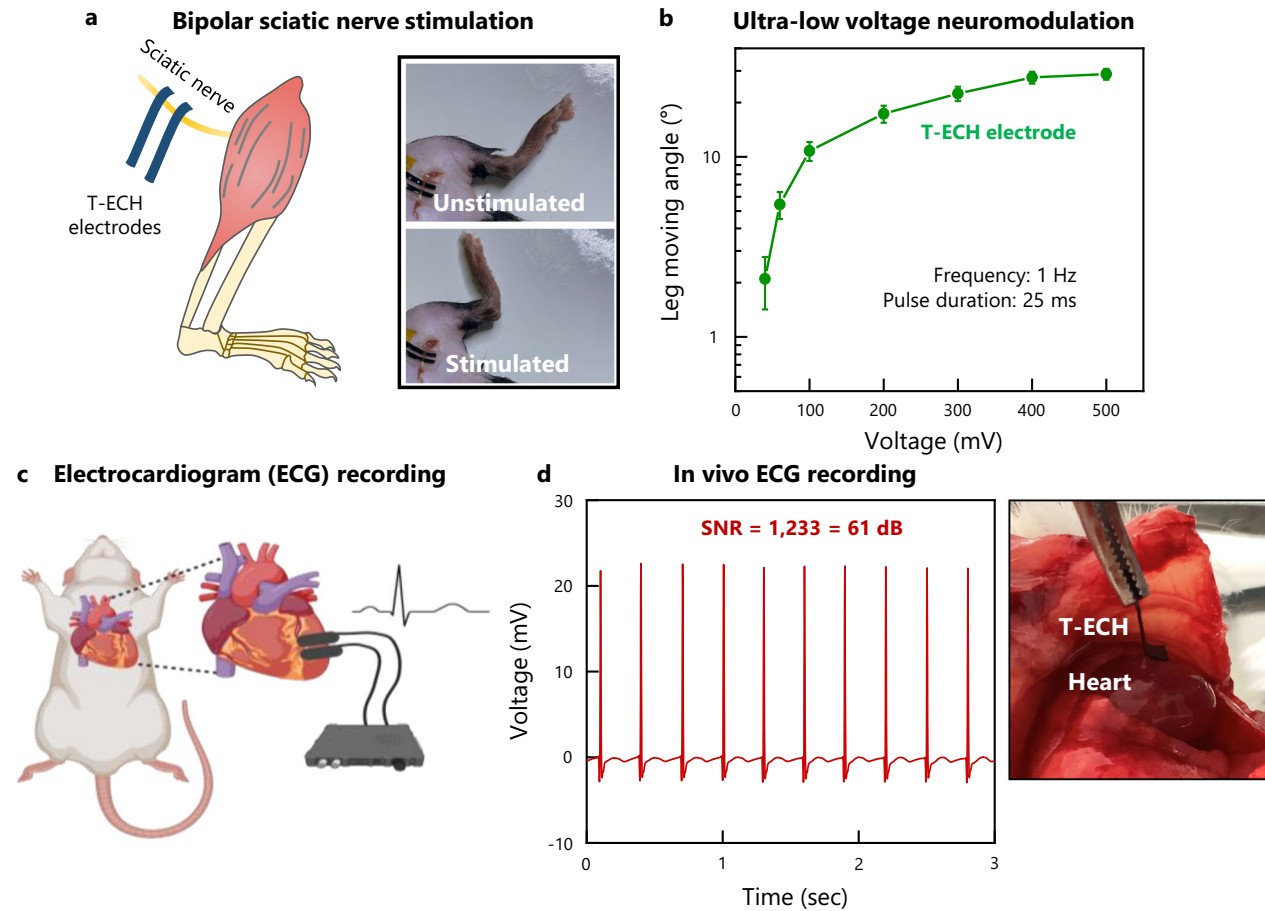

**Fig. 5 | In vivo bioelectronic device functionalities of T-ECH. a** Schematic illustration and image of bipolar sciatic nerve stimulation with T-ECH electrodes on the sciatic nerve. **b** Leg moving angle was measured for various stimulation voltages. Values represent the mean and the standard deviation (*n* = 9 devices examined with each mouse). The frequency was set to 1 Hz and the pulse duration was 25 ms. **c** Schematic illustration of in vivo ECG monitoring using T-ECH electrodes. Created with BioRender.com. **d** High-quality recording of ECG signals. SNR = 1122 (61 dB).

water content. Through this study, we demonstrated ultra-low voltage neuromodulation on the sciatic nerve and acquired high-quality ECG signals from heart tissues with high SNR in vivo animal models. The excellent biocompatibility, high electrical conductivity, tissue-like modulus, high toughness, and high-resolution patterning capabilities of T-ECH highlight its potential as a promising material in a variety of bioelectronic applications.

## Methods

### Materials
The chemicals purchased are used without further purification. Acrylic acid (99%), ammonium persulfate (APS) (98%), N N′-methylene bis(acrylamide) (MBAA) (99%), acrylamide (99%), dimethyl sulfoxide (DMSO) (99%), N-ethyl carbodiimide hydrochloride (EDC·HCl), and N-hydroxysuccinimide (NHS) (98%) were purchased from Sigma-Aldrich. PEDOT:PSS aqueous solution (Clevios™ PH1000, 1.0–1.3% solid content) was purchased from Heraeus Electronic Materials.

### Synthesis of electrically conductive hydrogel (ECH)
**T-ECH.** The acrylic acid solution is made by mixing acrylic acid monomers (576 mg, 8 mmol), APS (5.5 mg, 0.028 mmol), MBAA (4.3 mg, 0.024 mmol) in deionized (DI) water (2.76 mL). Then, PEDOT:PSS solution (5.24 mL, 0.1 g of PEDOT:PSS/1 g of acrylic acid monomers) was mixed with the acrylic acid solution. The total water content in the final solution was 8 mL. After degassing, the precursor solution was thermal-polymerized at 75 °C for 2 h to make Pre T-ECH.

Then, DMSO solution (4 mL, 62.5 v% in DI water) was poured into Pre T-ECH and annealed at 97 °C for 16 h. Finally, T-ECH (T-ECH 1) was made by swelling and washing in DI water. T-ECH 2 and T-ECH 3 are synthesized in the same way with different contents of MBAA and PEDOT:PSS (Supplementary Table 1).

**Pure ECH.** PEDOT:PSS solution and DMSO (13 v%) were mixed thoroughly for 1 h. Then it was dry-annealed at 60 °C overnight, annealed at 130 °C for 1.5 h, and swelled in DI water to make Pure ECH.

**PAAm T-ECH.** PAAm T-ECH was made with the same procedure as T-ECH, but the acrylic acid monomers were changed to acrylamide monomers.

**T-ECA.** T-ECA was made by activating the adhesion layer at the surface of T-ECH. An aqueous mixture of EDC (15 mg/mL) and NHS (23 mg/mL) was dropped on T-ECH to make T-ECA.

### Mechanical characterization
Tensile, compression, and fracture tests were measured with a universal testing machine (UTM) (Instron 68SC-1). Samples with dimensions of 5 × 10 mm (width × length), and 6 × 2 mm (radius × thickness) were measured for the tensile test and compression test, respectively. Strain rates for the tensile test and compression test were 50%/min and 0.2%/min. For the measurements of fracture toughness, samples of width 30 mm and length 10 mm were notched and stretched with a

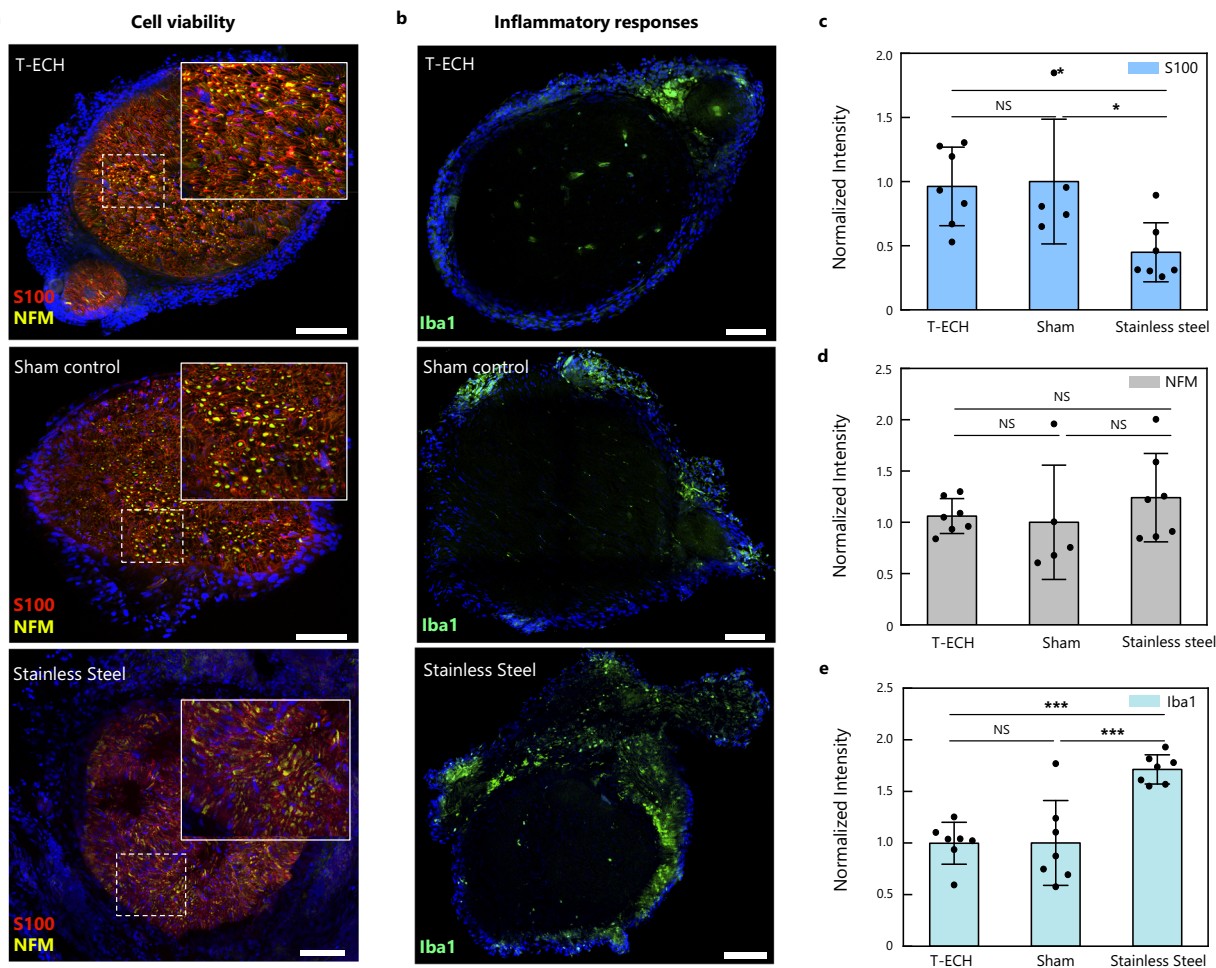

**Fig. 6 | Biocompatibility test of T-ECH. a**, **b** Representative immunofluorescence images of the sciatic nerve 14 days after implantation with T-ECH, sham group, and stainless steel (S-100 in red; NFM in yellow; Iba1 in green; DAPI in blue). Scale bar = 100 μm. **c**–**e** Normalized fluorescence intensity of T-ECH, stainless steel, and sham group respectively for (**c**) S-100, (**d**) NFM, and (**e**) Iba1. Data are plotted as mean values and the error bar represents the standard deviation. One-way ANOVA and Tukey's multiple comparison tests were employed for statistical analysis; *$P < 0.05$

***$P < 0.001$ with a confidence level of 95%. ($n = 5$ devices examined with individual mice for S100 and NFM of sham. $n = 7$ for S100 and NFM of T-ECH and stainless steel, $n = 7$ for Iba1 of all groups. For comparison of S-100 intensity, $P = 0.03$ for both between T-ECH and stainless steel and between Sham and stainless steel. For comparison of Iba1 intensity, $P = 0.0004$ for both between T-ECH and stainless steel and between Sham and stainless steel).

50%/min strain rate. Rheological properties were measured with an oscillatory rheometer (MCR302, Anton Paar) using 8 mm parallel plate-plate geometry. Frequency sweep measurements were conducted with 0.25 N force applied with 1% shear strain.

### Electrical characterization

Electrical conductivity was measured with a 4-point probe (M4P-205 System, MS TECH) connected to a source meter (2400 SourceMeter, Keithley). The conductivity was measured after swelling in D.I. water or PBS for 24 h. The measured T-ECH had a length of 10 mm and a width of 5 cm. The thickness of T-ECH 1 was 600 μm, T-ECH 2 was 800 μm, and T-ECH 3 was 1 mm. It was measured with the stainless-steel wire electrodes attached to the surface of the hydrogel and the wires were connected to the probe tips. For measuring the resistance of T-ECHs when stretching, hydrogel samples with 10 mm length and 5 mm width were attached to the stretcher and connected to the source meter with stainless steel wires. They stretched at strain rates of 200%/min and 300%/min for the tensile test and cyclic tensile test, respectively, with a bending & stretchable machine system (SnM). The resistance was simultaneously measured using a source meter (2400 SourceMeter, Keithley). Impedance and CV measurements of

T-ECHs and gold were performed by using a potentiostat (VSP-300, BioLogic) in PBS with a platinum electrode as the counter electrode and Ag/AgCl as the reference electrode. The exposed area of T-ECHs in PBS was kept at 3 mm × 3 mm. Potentiostatic electrochemical impedance spectroscopy (PEIS) was done with an amplitude of 500 mV, measured from 1 Hz to 1 MHz. For measuring the impedance of T-ECH when stretching, the impedance of T-ECH was measured with different strains in PBS with platinum as the counter electrode. CV was measured at a scan rate of 100 mV/s from −0.5 to 0.5 V. CSC was calculated from the CV curves according to the equation below.

$$CSC = \frac{\int (current\ density\ per\ area)\ dE}{2 \times (scan\ rate)}$$

### Small-angle neutron scattering (SANS), small-angle X-ray scattering (SAXS), and wide-angle X-ray scattering (WAXS)

SANS measurements of hydrogels were performed using a 40 m SANS instrument at HANARO, the Korea Atomic Energy Research Institute (KAERI) in Daejeon, Republic of Korea. The average

wavelength of the neutron beam was 6 Å. Two sample-to-detector distances (SDDs) of 2.5 m and 14.89 m were used to cover the q range. The hydrogels were swelled with deuterated water and put into the cells. The intensities were corrected with background scattering, empty cell scattering, and the sensitivity of detector pixels in deuterated water. Then, the absolute intensity was calculated by using the direct beam flux method. SAXS and WAXS measurements were performed at the 4 C SAXS II beamline at the Pohang Accelerator Laboratory (PAL) in Pohang, Republic of Korea. Storage ring energy was 3.0 GeV with an energy resolution ($\Delta E/E$) of $2 \times 10^{-4}$. The sample-to-detector distances (SDDs) were set at 4 m, 1 m, and 20 cm. SAXS measurement hydrogels were measured after swelling in DI water for 24 h and WAXS measurements were performed with fully dried films.

## UV-vis-NIR spectroscopy

T-ECH sample was prepared as follows. Acrylic acid monomer (100 mg) and APS (1 mg) were mixed in DI water (1 mL) to make a precursor solution. Then, the precursor solution was thermal-polymerized at 75 °C for 2 h to make the PAA solution. Then, diluted PAA solution (20 mg/mL) was mixed with PEDOT:PSS solution (10 mg/mL) and spin-coated on O$_2$ plasma-treated glass at 2000 rpm for 30 s. Then, it was annealed at 90 °C for 15 min, followed by dropping DMSO on the sample and annealing at 125 °C for 2 h. For pure PEDOT:PSS sample, PEDOT:PSS solution was spin-coated on O$_2$ plasma-treated glass at 2000 rpm for 30 s and annealed at 90 °C for 1 h. The glass substrates used were cleaned through sequential washing with acetone, ethanol, and DI water. UV-vis-NIR absorption spectra were measured using UV-Visible/NIR spectrophotometer (V-770, Jasco).

## X-ray photoelectron spectroscopy (XPS)

Fully dried films were measured for XPS (K-alpha, Thermo VG Scientific) with a monochromic Al K-alpha source under a high vacuum condition of $10^{-9}$ Torr.

## AFM, TEM, SEM, and micro-CT imaging

**AFM**. T-ECH sample was prepared as follows. Acrylic acid monomer (100 mg) and APS (1 mg) were mixed in DI water (1 mL) to make a precursor solution. Then, the precursor solution was thermal-polymerized at 75 °C for 2 h to make the PAA solution. Then, diluted PAA solution (20 mg/mL) was mixed with PEDOT:PSS solution (10 mg/mL) and spin-coated on an O$_2$ plasma-treated Si wafer at 2000 rpm for 30 s. Then, it was annealed at 90 °C for 15 min, followed by dropping DMSO on the sample and annealing it at 125 °C for 2 h. Pre T-ECH sample was prepared as follows. Acrylic acid monomer (100 mg) and APS (1 mg) were mixed in DI water (1 mL) to make a precursor solution. Then, the precursor solution was thermal-polymerized at 75 °C for 2 h to make the PAA solution. Then, diluted PAA solution (20 mg/mL) was mixed with PEDOT:PSS solution (10 mg/mL) and spin-coated on an O2 plasma-treated Si wafer at 2000 rpm for 30 s. Then, it was annealed at 90 °C for 3 h. For the Pure ECH sample, PEDOT:PSS solution was spin-coated on an O$_2$ plasma-treated Si wafer at 2000 rpm for 30 s and annealed at 90 °C for 15 min, followed by dropping DMSO on the sample and annealing it at 125 °C for 2 h. AFM images were obtained using AFM Microscope (NX10, Park Systems).

**TEM**. For PEDOT:PSS/PAA with DMSO sample, PEDOT:PSS solution was mixed with PAA (1:1) and DMSO (25 v%) was added. PEDOT:PSS with DMSO sample was made by adding DMSO (25 v%) in PEDOT:PSS solution. Then, all the samples were deposited on the TEM grid (Carbon Film Supported Copper Grid, Sigma-Aldrich) and dry-annealed. TEM measurement was done using an ultrahigh-resolution analytical electron microscope (JEM-3010, JEOL).

**SEM**. Hydrogels were freeze-dried by freezing into liquid nitrogen and drying in freeze-drier under vacuum conditions of $4 \times 10^{-3}$ Torr. Cross-sectional images were obtained using SEM (S-4800, Hitachi).

**Micro-CT**. Hydrogels were freeze-dried by freezing into liquid nitrogen and drying in freeze-drier under vacuum conditions of $4 \times 10^{-3}$ Torr. Micro-CT images were obtained using X-ray Micro Computed Tomography (SkyScan 1272, Bruker).

## Adhesion tests

All samples were prepared with a width of 2 cm and attached to PET backing using Krazy glue. Pig skin, artery, kidney, heart, and liver tissues were purchased from the local store. Adhesion tests were performed with a universal testing machine (UTM) (Instron 68SC-1). 180 °C peeling test and shear test were done with a strain rate of 50 mm/min. Interfacial toughness was obtained by dividing the force by the width of the sample. Shear stress was determined by dividing force by the attached area.

For the gold adhesion test, the gold substrate was functionalized with amine groups. First, the gold substrate was cleaned through acetone, ethanol, and DI water washing. Then, it was O$_2$ plasma cleaned with 50 W for 1 min. The cleaned gold was then immersed in 1 mM aqueous cysteamine solution for 1 h, followed by washing with DI water and drying under a vacuum.

## In vivo sciatic nerve stimulation

All animal experiments were conducted under the review and regulations of the Institutional Animal Care and Use Committee of KAIST. Male C57BL/6 N mice aged 8–10 weeks (Koatech, South Korea) were used in the sciatic nerve stimulation, and all the procedures were conducted under aseptic conditions. Mice were anesthetized with isoflurane (2–5% in oxygen), and the infrared lamp was given to maintain the body temperature. After shaving the leg hairs, a 1 cm incision was made on the medial hindlimb. The sciatic nerve was exposed by slitting the biceps femoris muscle. T-ECH electrodes for nerve stimulation were prepared in size 1 mm × 3 mm, and back-insulated with PDMS-based self-healing polymer. Two electrodes were gently placed under the sciatic nerve. The biphasic pulses were given with an isolated pulse stimulator (2100, A-M systems). The pulse parameter of frequency and pulse duration was set to 1 Hz and 25 ms, respectively. The voltage for stimulation ranged from 40 to 500 mV. The angle of leg movement was measured with a protractor placed beneath the leg. For precise comparison, the leg was in its original position for each measurement.

## In vivo ECG recording

Female SD rats aged 10–12 weeks (Koatech, South Korea) were used in acute epicardial recording experiments. Rats were anesthetized with isoflurane (2–5% in oxygen) for general anesthesia. An acute tracheotomy was conducted and the rats were intubated on a ventilator (VentElite Small Animal Ventilator, Harvard Apparatus) at 90 breaths per minute. The ribs were cut to expose the heart and the pericardium was gently removed. Electrodes were prepared of size 1 mm × 3 mm and connected to silver paint to copper wires. Electrodes were placed 2 mm apart with a PDMS-based self-healing polymer. During ECG recording, working and reference electrodes were placed on the left atrium and recorded with a bandpass filter of 0.3 Hz -70 Hz (Lab Rat System, Tucker-Davis Technologies). Both electrodes were lightly pressed for 5 s to enable robust adhesion on the epicardial surface.

## Immunohistochemistry (IHC)

Male mice aged 8–10 weeks were used for immunohistochemistry, and their sciatic nerves were exposed as described in the In-vivo sciatic nerve stimulation section. T-ECH electrodes and stainless-steel films with a size of 1 mm × 3 mm were implanted around the sciatic nerve

after being sterilized with 70% ethyl alcohol and 1 h of UVC illumination. For the sham group, surgery to expose sciatic nerve was conducted without the implant. For chronic implantation without infection, the incision was sutured with bioabsorbable VICRYL sutures (Ethicon), and antiseptic was applied. Following recovery, mice were housed individually and maintained at 22 ˚C, 12-h light/dark cycle, and 45% humidity with free access to the food and water ad libitum. After 2 weeks of implantation, mice were transcardially perfused with 4% paraformaldehyde (PFA) in PBS. The sciatic nerves were dissected, kept in 4% PFA for 16 h at 4 ˚C, and transferred in 30% sucrose solution at 4 ˚C for cryoprotection. The nerves, frozen within the OCT compound (Tissue-Tek OCT compound, Sakura), were sectioned for staining. The sections were blocked and permeabilized in 0.3% Triton-X 100 (Sigma) and 5% normal donkey serum (Abcam) in PBS. After rinsed in PBS, the sections were incubated overnight at 4˚C with primary antibodies; Anti-S100 (1:200, ab52642, Abcam) and Anti-160kD NFM (1:200, ab195658, Abcam), or Anti-Iba1 (1:1000, ab107159, Abcam). The sections were rinsed with PBS and incubated for 2 h with secondary antibodies Donkey Anti-Goat IgG H&L Alexa Fluor 488 (1:250, ab150129, Abcam) and/or Donkey Anti-Rabbit IgG H&L Alexa Fluor 594 (1:250, ab150076, Abcam). The sections were mounted and counterstained with DAPI, 4'6-diamidino-2-phenylindole (Vectashield, Vector Laboratories). Sections were examined with a laser-scanning confocal microscope (C2, Nikon) for image acquisition. The quantification of fluorescent intensity was analyzed with ImageJ by cropping the contour of the sciatic nerve and averaging the intensity values using the same software.

### Histology (hematoxylin and eosin stain)

Implants in male mice were conducted as described in the Immunohistochemistry section. After a 2-week implantation period, the mice were humanely euthanized by $CO_2$ inhalation. The sciatic nerve and surrounding muscle tissue were carefully dissected. The tissues of interest were then fixed in 10% formalin for 24 h prior to histological processing. The fixed tissues were sequentially dehydrated in increasing concentrations of ethanol before being embedded in paraffin with a tissue processor (Leica TP1020). The embedded tissues were then sectioned with a rotary microtome (Leica RM2235) and stained with Hematoxylin and Eosin for microscopic examination. The blind histological evaluation was conducted by a histopathologist.

### Statistical analysis

GraphPad Prism (GraphPad Software) was used for statistical analysis. For immunohistochemistry, the normality of the fluorescence intensity data was assessed using the Kolmogorov-Smirnov test, while the homogeneity of variance was evaluated using the Brown-Forsythe test. One-way ANOVA and Tukey's multiple comparison test were employed to assess the statistical significance with threshold of $^*P < 0.05$, $^{**}P < 0.01$, $^{***}P < 0.001$.

### Reporting summary

Further information on research design is available in the Nature Portfolio Reporting Summary linked to this article.

## Data availability

The data that support the findings of this study are available within this article and its Supplementary Information. Source data is provided as Source Data file. Data is also available from the corresponding author upon request. Source data are provided with this paper.

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

## Acknowledgements

This study was supported by the National R&D Program through the National Research Foundation of Korea (NRF) funded by the Ministry of Science and ICT (2021M3H4A1A04092882 and 2022M3H4A1A04096393). Figure 5c was created with BioRender.com.

## Author contributions

J.C., S.C., S.P., and J.K. conceived and designed the experiments. J.C. synthesized and characterized T-ECH. J.C., T.K., H.P., and J.K. analyzed the data. J.C. and H.K. performed mechanical tests. J.C., C.S., and K.S.N. performed in vivo experiments. C.S. performed immunofluorescence staining and data analyses. J.C. and H.L. obtained AFM data. H.P. synthesized self-healing polymer backing. J.C., C.S., K.S.N., S.P., and J.K. wrote the paper. All the authors discussed the results and commented on the manuscript. J.K. supervised the study.

## Competing interests

The authors declare no competing interest.
