## [Peer Review File · Nature Communications]

Highly conductive tissue-like hydrogel interface through template-directed assemblyReviewers' Comments:

Reviewer #1:

Remarks to the Author:

Chong and co-authors developed a highly conductive hydrogel with poly(3,4-ethylenedioxythiophene): poly(styrene sulfonate) (PEDOT:PSS) and polyacrylic acid (PAA) by utilizing a templated-directed assembly method. The resulting hydrogel composite features high mechanical toughness, tissue-like modulus, and tough adhesion. The authors used the PAA polymeric network as a soft template to form multivalent hydrogen bonds with the PSS shell, allowing the ultrathin PEDOT:PSS fibrous network to grow along the template when the solvent (DMSO) is introduced. Followed by the dry-annealing and re-swelling process, the PEDOT network was connected to form an electrical pathway through the PAA template chains. This assembly method made PEDOT:PSS not to form large domains of bulk aggregates but to form a nanofibrous conductive network in-between the PAA network, enabling the hydrogel composite to be highly conductive and stretchable at the same time. The characteristic properties of the proposed hydrogel were explored well through neuromodulation and epicardial electrocardiogram (ECG) recording demonstrations. This work is a meaningful contribution for the progress of the field of soft bioelectronics, and thus the reviewer recommends publication of this manuscript in Nature Communications after addressing following issues through a proper revision.

Comment #1: When a PAA template network is produced (Pre T-ECH), PEDOT:PSS initially appears in a colloidal form, whereas it develops into extended nanofibers after dry-annealing with DMSO. The authors need to show the conformational change to prove the mechanism of the improved conductivity. Please provide additional data of Pre T-ECH (e.g., one of SANS, AFM, XPS, and WAXS analysis) with the T-ECH.

Comment #2: In supplementary figure 4, the authors demonstrated T-ECH using polyacrylamide (PAAm) and poly(2-hydroxyethyl methacrylate) (pHEMA). In figure S4a and S4b, stress-strain curve of polyHEMA T-ECH and image of PAAm T-ECH should be added to show low fracture strain and phase segregation in both experimental groups. In figure S4c, it is hard to distinguish phase-separated domains of PEDOT:PSS and PAAm. Please inform how this image was obtained and indicate each domain for better understanding. Also, it seems that figure S4d is not mentioned anywhere in the main text.

Comment #3: In figure 2a, the authors depicted 'Template network' with an arrow between T-ECH and Pure ECH. This causes misunderstanding, such as Pure ECH is fabricated from T-ECH. Please revise figure 2a more clearly.

Comment #4: In figure 2f, the authors described that the T-ECH electrode exhibited a high conductivity (247 S/cm) even in the presence of 90 wt% water content and 9 wt% nonconductive PAA content. Authors should provide detailed information of the conductivity measurement condition (e.g., swelling time, composition of T-ECH).

Comment #5: In figure 3g, the reviewer thinks conductivity is plotted incorrectly. According to the Table S1, T-ECH 1 has conductivity of 24765 S/m and 400% of strain at break, and T-ECH 2 has conductivity of 934 S/m and 610% of strain at break. However, the conductivity of two points of this work does not match. Please indicate exactly which sample is the corresponding point. Additionally, the boundary that depicts 'this work' is ambiguous. Authors should delete the boundary of the colored area.

Comment #6: In page 16, the authors explained that rigid cuff devices induce increased fibrosis and myelinated fiber density. From figure 6, it is difficult to observe whether fibrosis occurred or not. Please provide the data for fibrosis assessment, such as H&E-stained image, and the myelinated fiber density.

Reviewer #2:

Remarks to the Author:

Comment:

1. Bioelectronics field has been long waiting for a robust, highly stretchable, low impedance and adhesive electrode materials, which are critical for bioelectronic interface to electrogenic tissues for bidirectional and chronically stable electrophysiological communication. It was a big challenge to realize all those properties at the same time. Dr. Kang and Dr. Park's group successfully addressed this daunting challenge and developed an electrically conductive hydrogel through template-directed assembly (termed T-ECH). The approach to synthesis T-ECH is innovative and interesting. The adhesive hydrogel material realized both high electrical conductance of over 100 S/cm and stretchability of over 200% strain with tissue-like mechanical properties. Those are significantly better than state-of-the-art ECH materials. The authors subsequently demonstrated the application for both electrophysiological recording and stimulation and achieved lowest stimulation voltage to elicit leg movement through sciatic nerve stimulation. I believe the T-ECH can be widely used for all kinds of bioelectronic interface considering its superior electrical and mechanical properties and will be of great interest for anyone working in bioelectronics and of great relevance to the broader bioengineering and materials field. This work is very well done, and I recommend to accept this manuscript after addressing the comments below.

2. The authors refer to the newly developed hydrogel materials as "adhesive-free" bioelectronic. This term "adhesive-free" is a little bit confusing because the material is an adhesive hydrogel. The authors can consider change this term.

3. The authors proposed the structure of flattened, quasimetallic PEDOT-rich grains are organized in horizontal layers that are separated by continuous insulating PSS-PAA lamellas. Please characterize and explain the anisotropic mechanical (e.g., Young's modulus in out of plan direction) and electrical (e.g., conductivity in out of plan direction) properties, as it is important and unique to the reported conductive hydrogel materials. Please elaborate on the nearest-neighbor hopping across the PSS-PAA lamellas and how does that compare with pure electrically conductive hydrogel previously reported?

4. Based on previous research works on PEDOT:PSS (e.g., J. Rivnay Nat. Commun.2016, 7, 11287.) the hydrogel is expected to have a PEDOT:PSS-rich fiber (towards the ultimate limit of 1:1 PEDOT:PSS), and PSS-PAA lamellas. Please consider to change the term "PEDOT fiber" to "PEDOT:PSS fiber" as PEDOT is doped by PSS, and modify the schematic drawing on Fig 3a.

5. The author explained that continuous breakage/reformation of hydrogen bonds between PEDOT:PSS network and PAA template contributes to the stretchability. Please characterize or explain the viscoelasticity or viscoplasticity of the materials. To what degree the deformation is irreversible? How does the stress-strain curve response to different strain rate.

Minor Points:

1. How does dry-annealed at 95 °C improve the conductivity and mechanical properties? Can you explain the rationale for dry-annealing?

2. "T-ECH can endure a high compressive force (35 N)": I suggest the authors to use areal normalized parameter, i.e., stress instead of force to describe the mechanical properties.

3. Referring to the sentence "Therefore, the fine extended thin PEDOT fibers have fast and efficient

charge transport via intra-chain transport, which is faster than inter-chain transport (Supplementary Fig. 8 and 9)", only supplementary Fig. 8 illustrate the inter-chain transport. Fig S9 is rheology comparison between T-ECHs and Pure ECH. The authors should add explanation about how lower degree of entanglements from Fig. S9 contributes to more intra-chain transport.

4. There is a typo in Fig S10 caption "Therefore, we can conclude that PEDOT in T-ECH is in linear and thin fibrous network, network," There are two "network"s

5. There is a typo in the following sentence "Compared with PEDOT:PSS, T-ECH had a drastically increased ration of PEDOT to PSS from 0.53 to 0.88". Ration > Ratio

Reviewer #3:

Remarks to the Author:

Chong et al here report on a type of electrically conducting hydrogel comprised of Polyacrylic Acid and nanostructured PEDOT:PSS. With their method, a highly stretchable and electrically conductive hydrogel is achieved, that finds potential use in bioelectronic applications.

The work is based on their previously reported nano-confinement method for obtaining a well-ordered PEDOT network. This has now been demonstrated to also work within a PAA hydrogel. With this, conductivities on the order of 240 S/cm were achieved, which are very high for polymer-based hydrogel composites.

However, several issues exist with the current version of the manuscript.

For once, the title prominently suggests the existence of a metallic nano fibrous network, where in reality a conductive polymer based network is used. This is reasoned later on in the text with the measured conductivity, and the relatively low mass percentage of PEDOT:PSS that is hypothesized to imply metallic conductivity in said PEDOT. No direct proof of that hypothesis is given. Then, quite misleadingly, an ashby plot (Figure 3 g) is labeled "Ashby Plot of conducting hydrogels", where in reality only polymer-conductor-based hydrogels are shown in that plot (an information stated in the caption, but that is really misleading). Consequently, stretchable conducting hydrogels based i.e. on (actually metallic) silver nanostructures are omitted from the plot. Such networks exhibit even higher conductivity (>350 S/cm, in comparison to the here reported 247 S/cm).

This is even more surprising, as the corresponding work is then found as reference in the supplemental material (1. Ohm, Y. et al. An electrically conductive silver-polyacrylamide-alginate hydrogel composite for soft electronics. *Nat. Electron.* 4, 185–192 (2021)). Now, talking about metallic nano fibrous networks in the title, but then leaving out actual metal-based conducting hydrogels in the comparison is highly dubious. If such comparison plots are included, they should be balanced and fair. Metallic conductivity in the polymer network should be unambiguously proven if claimed. In the methods, the authors mention the use of 4-point probes (M4P-205 System, MS TECH) that press onto the hydrogel (with probe heads very close to each other) to measure conductivity. Specifics on that setup would be helpful. They are typically used to measure sheet resistance in thin solid films. Is the hydrogel compressed during measurement? conductivities in excess of 2400 S/m in PEDOT:PSS thin films have been reported. Such a setup seems not very suitable to measure conductivity in large, soft sheets of hydrogels, there may be a source of significant error here.

Related, the key figures of merit (conductivity, toughness) lack error bars, the statistics is unclear (Fig 2f).

Then, the term "Adhesive-free" in the title implies that no further treatment of the hydrogel is needed in order for it to adhere to various surfaces. This is contradicted by the need of an "activation layer"

(Figure 4a, methods: "An aqueous mixture of EDC (15 mg/mL) and NHS (23 mg/mL) was dropped on T-ECH to make T-ECA", gold surfaces had to be functionalized as well to allow for adhesion). Now, it is perfectly fine to do such modifications to achieve tough adhesion. But "adhesive-free" should then not be used, because it is again misleading.

The mechanical data on the cyclic fatigue tests should be shown as well, not only the electrical one. I wonder if the nano-structured PEDOT network fatigues. Corresponding SEM images (as done in figure S14) after mechanical fatigue might also be helpful here.

Figure S6 d-f, what are the values for the initial resistance in those cases? (in Ohms, not the normalized values). What are the dimensions of the samples? Again, more information on how conductivity was actually measured would be very valuable.

In conclusion, this work needs at least mayor revisions. The measured conductivity values have to be unambiguously proven, and correctly benchmarked to conducting hydrogels at large.

Point-to-Point **Responses** to Reviewers' Comments

Reviewer #1

Chong and co-authors developed a highly conductive hydrogel with poly(3,4-ethylenedioxythiophene): poly (styrene sulfonate) (PEDOT:PSS) and polyacrylic acid (PAA) by utilizing a templated-directed assembly method. The resulting hydrogel composite features high mechanical toughness, tissue-like modulus, and tough adhesion. The authors used the PAA polymeric network as a soft template to form multivalent hydrogen bonds with the PSS shell, allowing the ultrathin PEDOT:PSS fibrous network to grow along the template when the solvent (DMSO) is introduced. Followed by the dry-annealing and re-swelling process, the PEDOT network was connected to form an electrical pathway through the PAA template chains. This assembly method made PEDOT:PSS not to form large domains of bulk aggregates but to form a nanofibrous conductive network in-between the PAA network, enabling the hydrogel composite to be highly conductive and stretchable at the same time. The characteristic properties of the proposed hydrogel were explored well through neuromodulation and epicardial electrocardiogram (ECG) recording demonstrations. This work is a meaningful contribution for the progress of the field of soft bioelectronics, and thus the reviewer recommends publication of this manuscript in Nature Communications after addressing following issues through a proper revision.

=> We appreciate these highly encouraging comments.

=> We have now carefully revised the manuscript according to the reviewer's comments and suggestions. All main revisions are marked in red font.

Comment #1: When a PAA template network is produced (Pre T-ECH), PEDOT:PSS initially appears in a colloidal form, whereas it develops into extended nanofibers after dry-annealing with DMSO. The authors need to show the conformational change to prove the mechanism of the improved conductivity. Please provide additional data of Pre T-ECH (e.g., one of SANS, AFM, XPS, and WAXS analysis) with the T-ECH.

=> We thank the reviewer for this suggestion and comment. In order to investigate the conformational changes of PEDOT:PSS, we conducted small angle X-ray scattering (SAXS) experiments of PAA, Pre T-ECH, and T-ECH. It was seen that Pre T-ECH and PAA have similar features, meaning that there is no significant conformational difference between Pre T-ECH and PAA (amorphous). This result explains that colloidal PEDOT:PSS in Pre T-ECH doesn't have a long-range fibrous network. However, T-ECH showed a long-range fibrous PEDOT:PSS network structure inside the PAA template network. Specifically, T-ECH showed a higher slope at a low q range ($q < 0.01 \text{ \AA}^{-1}$) than Pre T-ECH and PAA, indicating a needle-like PEDOT network is made inside PAA. At a mid q range ($q > 0.1 \text{ \AA}^{-1}$), T-ECH had a lower slope than Pre T-ECH and PAA due to the linear structure of PEDOT chains made in complex and flexible PAA chains. Therefore, through DMSO annealing and re-swelling, PEDOT:PSS colloids in Pre T-ECH transform into the fibrous network with linearly extended PEDOT chains in T-ECH. Consequently, when PEDOT:PSS network has been made through DMSO annealing, it shows high electrical conductivity due to the linear and thin PEDOT:PSS fiber formation inside the template polymer.

Fig. S12| SAXS analysis of Pre T-ECH and T-ECH.

Hydrogels were measured after swelling in DI water for 24 hours. The SAXS measurement was performed at the 4C SAXS II beamline at the Pohang Accelerator Laboratory (PAL) in Pohang, Republic of Korea. Storage ring energy was 3.0 GeV with an energy resolution ($\Delta E/E$) of 2×10^{-4} . The sample-to-detector distances (SDDs) were set at 4 m, 1 m, and 20 cm to cover the q range.

Main text, Page 10-11

In addition, to obtain conformational change from Pre T-ECH to T-ECH, small-angle X-ray scattering (SAXS) was conducted (Supplementary Fig. 12). It was found that DMSO annealing and re-swelling transformed colloidal PEDOT:PSS to thin fibrous PEDOT:PSS network, making ordered PEDOT:PSS/PAA lamella structure.

=> Additionally, clear conformational changes of PEDOT:PSS were confirmed by atomic force microscopy (AFM). The DMSO annealing process transformed the colloidal PEDOT:PSS into a thin fibrous PEDOT:PSS network. Pre T-ECH doesn't have any long-range assembly of PEDOT:PSS while T-ECH has a fibrous long-range network.

Supplementary Figure 13| AFM height and amplitude images of Pre T-ECH and T-ECH.

a, b, Height images of Pre T-ECH, T-ECH. **c, d,** Amplitude images of Pre T-ECH, T-ECH. T-ECH has a nanofibrous network of PEDOT:PSS whereas no such fibrous network is seen in Pre T-ECH. AFM images were obtained using AFM Microscope (NX10, Park Systems). More detailed information about the AFM samples is in the method section (Main text, Page 28-29).

Main text, Page 10-11

It was found that DMSO annealing and re-swelling transformed colloidal PEDOT:PSS to thin fibrous PEDOT:PSS network, making ordered PEDOT:PSS/PAA lamella structure. This conformational change was further seen in the atomic force microscopy (AFM) image (Supplementary Fig. 13), confirming that PEDOT:PSS fibers are made in T-ECH.

=> To further investigate the effect of DMSO annealing, X-ray photoemission spectroscopy (XPS) experiment was conducted. The XPS spectra of T-ECH showed clear peaks at

approximately 167 and 162.5 eV due to the S 2p spectra of PSS and PEDOT moieties, respectively. In Pre T-ECH, PEDOT:PSS exists in colloidal aggregates without making a fibrous network. However, in T-ECH, PEDOT and PSS form a thin fibrous network along the template network. The thin PEDOT fibers in T-ECH have a more accessible 3-dimensional area than the colloidal PEDOT structure in Pre T-ECH, resulting in more interaction with PSS polyanions. Therefore, the S 2p peak of PEDOT shifts to lower energy. Similarly, PSS can interact more effectively with PAA when they have a structure of a fibrous thin network rather than a bulk aggregate. Therefore, the S 2p peak of PSS in T-ECH shifted to lower energy compared to the Pre T-ECH. In addition, the S 2p peak of PSS in Pre T-ECH slightly shifted to lower energy than PEDOT:PSS, meaning that PSS forms hydrogen bonding with the PAA template.

Supplementary Figure 19| XPS spectra of the fully dried films of Pure ECH, Pre T-ECH, and T-ECH. Fully dried films were measured for XPS (K-alpha, Thermo VG Scientific) with a monochromic Al K-alpha source under a high vacuum condition of 10^{-9} Torr.

Main text, Page 12

In addition, the structural difference of Pre T-ECH and T-ECH was investigated through XPS analysis (Supplementary Fig. 19). It was found that Pre T-ECH had colloidal bulk PEDOT:PSS.

Therefore, it was confirmed that the DMSO annealing process made the fibrous PEDOT:PSS network inside PAA.

Comment #2: In supplementary figure 4, the authors demonstrated T-ECH using polyacrylamide (PAAm) and poly(2-hydroxyethyl methacrylate) (pHEMA). In figure S4a and S4b, stress-strain curve of polyHEMA T-ECH and image of PAAm T-ECH should be added to show low fracture strain and phase segregation in both experimental groups. In figure S4c, it is hard to distinguish phase-separated domains of PEDOT:PSS and PAAm. Please inform how this image was obtained and indicate each domain for better understanding. Also, it seems that figure S4d is not mentioned anywhere in the main text.

=> We thank the reviewer for this suggestion and comment. In the case of polyHEMA T-ECH (Fig S4c), it was not able to perform mechanical tests due to its poor mechanical properties. This phenomenon results from the severe phase separations of PEDOT:PSS and polyHEMA due to the low binding affinities between PEDOT:PSS and polyHEMA. We have added this detail in Fig S4b for more information about polyHEMA T-ECH.

=> Photo of PAAm T-ECH was taken with a flashlight. The bright areas are PAAm and the dark areas are PEDOT:PSS domains. Still, we observed the phase separations of PEDOT:PSS and PAAm in the hydrogel. The non-uniform network structure of PAAm T-ECH results in low stretchability, and relatively low electrical conductivity. We specified those domains and give more description about the condition of the photography in Fig S4. Also, Fig S4d is moved to Fig 5c and mentioned in the main text.

Fig S4| T-ECH using PAAm and PolyHEMA as the template polymer.

a, Stress-strain curve of PAAm T-ECH 2, with 0.5 mol% of crosslinker and 5 wt% of PEDOT:PSS. The electrical conductivity of PAAm T-ECH 2 is 0.01 S/cm. **b**, Image of PAAm T-ECH taken under the flashlight. PAAm T-ECH shows phase-separated domains of PEDOT:PSS and PAAm (Bright area: PAAm, dark area: PEDOT:PSS). Scale bar = 3 mm. **c**, Image of poly(2-hydroxyethyl methacrylate) (polyHEMA) T-ECH. Due to the weak interaction between polyHEMA and PEDOT:PSS, there is a high energetic disagreement between PEDOT:PSS and polyHEMA. Consequently, it shows significant phase separation of PEDOT:PSS and polyHEMA and couldn't be developed into a hydrogel. Scale bar = 1 cm.

Fig. S5c| Image of measuring the resistance of T-ECH 1 with a multimeter. Scale bar = 2 cm.

Main text, Page 8

Therefore, the electrical resistance of T-ECH measured was significantly low (Supplementary Fig. 5c).

Comment #3: In figure 2a, the authors depicted ‘-Template network’ with an arrow between T-ECH and Pure ECH. This causes misunderstanding, such as Pure ECH is fabricated from T-ECH. Please revise figure 2a more clearly.

=> We thank the reviewer for this suggestion and comment. We modified figure 2a to avoid misunderstanding about Pure ECH.

Fig. 2a| Schematic illustrating of Pre T-ECH, T-ECH and Pure ECH. T-ECH has a double network structure of template polymer (PAA) and PEDOT:PSS. Pre T-ECH lacks PEDOT:PSS network while Pure ECH lacks a template network.

Comment #4: In figure 2f, the authors described that the T-ECH electrode exhibited a high conductivity (247 S/cm) even in the presence of 90 wt% water content and 9 wt% nonconductive PAA content. Authors should provide detailed information of the conductivity measurement condition (e.g., swelling time, the composition of T-ECH).

=> We thank the reviewer for this suggestion and comment. For the conductivity measurement with the 4-point probe, we added more specific information in the method section. PEDOT:PSS hydrogels were cut into a rectangular shape (10 mm in length, and width of 5 mm in width). It was measured with the stainless-steel wire electrodes attached to the surface of the hydrogel without pressing and connected to the probe tips.

Fig. r1| Electrical conductivity measurement setup.

Main text, Page 25-26

Electrical characterization

Electrical conductivity was measured with a 4-point probe (M4P-205 System, MS TECH) connected to a source meter (2400 SourceMeter, Keithley). The conductivity was measured after swelling in D.I. water or PBS for 24h. The measured T-ECH had a length of 10 mm a width of 5 cm. The thickness of T-ECH 1 was 600 μm, T-ECH 2 was 800 μm, and T-ECH 3 was 1 mm. It was measured with the stainless-steel wire electrodes attached to the surface of the hydrogel and the wires were connected to the probe tips.

Comment #5: In figure 3g, the reviewer thinks conductivity is plotted incorrectly. According to the Table S1, T-ECH 1 has conductivity of 24765 S/m and 400% of strain at break, and T-ECH 2 has conductivity of 934 S/m and 610% of strain at break. However, the conductivity of two points of this work does not match. Please indicate exactly which sample is the corresponding point. Additionally, the boundary that depicts 'this work' is ambiguous. Authors should delete the boundary of the colored area.

=> We thank the reviewer for this suggestion and comment. The conductivity units were wrong in the original Fig. 3g. We corrected $S\ m^{-1}$ to $S\ cm^{-1}$. Also, we indicated the details of the sample (T-ECH 1, T-ECH 2) at the points.

Fig. 3g| Ashby plot of conductive polymer hydrogels. PEDOT:PSS-based hydrogel (blue), polyaniline (PANI)-based hydrogel (yellow), polypyrrole (PPy)-based hydrogel (green), and metal-based hydrogel (gray) (Supplementary Fig. 1).

Comment #6: In page 16, the authors explained that rigid cuff devices induce increased fibrosis and myelinated fiber density. From figure 6, it is difficult to observe whether fibrosis occurred or not. Please provide the data for fibrosis assessment, such as H&E-stained image, and the

myelinated fiber density.

=> We thank the reviewer for this great suggestion and comment. To clarify the ambiguity regarding the occurrence of fibrosis, we added biocompatibility tests with H&E staining in the revised manuscript. In the comparison between the T-ECH electrode implant group, sham group, and rigid stainless steel (SS) implant group, we confirmed that SS group exhibited a much higher degree of fibrosis compared to other the two groups (Supplementary Figure 23). In addition to that, we also added a rigid SS cuff group in the IHC experiments for further validation of increased fibrosis in stiff implants. The results demonstrate that the fluorescence intensity of S-100, the marker of myelin-forming Schwann cell was significantly decreased in the SS group. In addition, Iba1, a marker for activated macrophages, was significantly increased in the SS group, which is in agreement with previous reports (*Nat. Rev. Mater.* **3**, 17076 (2017), *J. Biomed. Mater. Res.* **50**, 215-226 (2000), and *J. Neuroimmunol.* **159**, 75-86 (2005)) (Figure 6).

Fig.6: Biocompatibility test of T-ECH. **a, b,** Representative immunofluorescence images of the sciatic nerve 14 days after implantation with T-ECH, sham group, and stainless steel (S-100 in red; NFM in yellow; Iba1 in green; DAPI in blue). Scale bar = 100 μ m. **c, d, e,** Normalized fluorescence intensity of T-ECH, stainless steel, and sham group respectively for (c) S-100, (d) NFM, and (e) Iba1. Error bar represents the standard deviation. One-way ANOVA and Tukey's multiple comparison test was employed for statistical analysis; * $P < 0.05$ *** $P < 0.001$. (n=5 for S100 and NFM of sham. n=7 for S100 and NFM of T-ECH and stainless steel, n=7 for Iba1 of all groups).

Supplementary Figure 23. Representative histological images of a, T-ECH electrode b, sham group c, stainless steel after 2 weeks of implantation. Scale bar = 500 μ m. The highest degree of fibrosis was observed in the histology of the stainless steel.

Main text, Page 17

Biocompatibility

To assess the potential for long-term bioelectronic interfaces, we evaluate the *in vivo* biocompatibility of T-ECH-based implants by wrapping bare T-ECH film around the sciatic nerve of mice. To assess the biocompatibility of the T-ECH, two control groups were tested. The same surgical procedure without the implantation of T-ECH electrodes was conducted on a sham group and the second group received rigid stainless steel (SS) film implants. After 2 weeks of implantation, the immune response and variation in the cell population of the sciatic nerve tissue were observed by immunohistochemistry (IHC) with the following markers; S-100 for Schwann cells, neurofilament medium (NFM) for neurofilaments (axons), and Iba-1 for macrophages (Fig. 6). The T-ECH implants exhibited no significant difference in the population of Schwann cells and axons compared to the sham group. In contrast, the rigid SS implant group demonstrated a significantly decreased population of myelin-forming Schwann cells due to the compression and occlusion by the pressure of the rigid substrate⁴⁸ (Fig. 6a, c, d). Moreover, there was no significant difference in the expression of Iba-1, a key indicator of

immune response, between the T-ECH and sham groups (Fig. 6b, e), whereas rigid SS implants evoked an increased immune response compared with the sham and T-ECH groups in consistency with previous reports^{6,49,50}. Similarly, the highest degree of fibrosis was observed in the rigid SS group with hematoxylin and eosin (H&E) staining (Supplementary Fig. 23)

Reviewer #2

Comment:

1. Bioelectronics field has been long waiting for a robust, highly stretchable, low impedance and adhesive electrode materials, which are critical for bioelectronic interface to electrogenic tissues for bidirectional and chronically stable electrophysiological communication. It was a big challenge to realize all those properties at the same time. Dr. Kang and Dr. Park's group successfully addressed this daunting challenge and developed an electrically conductive hydrogel through template-directed assembly (termed T-ECH). The approach to synthesis T-ECH is innovative and interesting. The adhesive hydrogel material realized both high electrical conductance of over 100 S/cm and stretchability of over 200% strain with tissue-like mechanical properties. Those are significantly better than state-of-the-art ECH materials. The authors subsequently demonstrated the application for both electrophysiological recording and stimulation and achieved lowest stimulation voltage to elicit leg movement through sciatic nerve stimulation. I believe the T-ECH can be widely used for all kinds of bioelectronic interface considering its superior electrical and mechanical properties and will be of great interest for anyone working in bioelectronics and of great relevance to the broader bioengineering and materials field. This work is very well done, and I recommend to accept this manuscript after addressing the comments below.

=> We appreciate these highly encouraging comments.

=> We have now carefully revised the manuscript according to the reviewer's comments and suggestions. All main revisions are marked in red font.

2. The authors refer to the newly developed hydrogel materials as “adhesive-free” bioelectronic.

This term “adhesive-free” is a little bit confusing because the material is an adhesive hydrogel. The authors can consider change this term.

=> We thank the reviewer for this suggestion and comment. We deleted “adhesive-free” in the title. The title is changed to “Highly conductive tissue-like hydrogel interface through template-directed assembly”.

3. The authors proposed the structure of flattened, quasimetallic PEDOT-rich grains are organized in horizontal layers that are separated by continuous insulating PSS-PAA lamellas. Please characterize and explain the anisotropic mechanical (e.g., Young’s modulus in out of plan direction) and electrical (e.g., conductivity in out of plane direction) properties, as it is important and unique to the reported conductive hydrogel materials.

=> We thank the reviewer for this suggestion and comment.

=> The PEDOT fiber network forms mainly in a horizontal direction due to the anisotropic evaporation of the DMSO solvent. However, although the conductive pathway favors forming in an in-plane direction, there are some PEDOT fibers that align in an out-of-plane direction. Since PEDOT:PSS fibers form through an isotropic 3D PAA template, we believe that some highly conductive PEDOT fibers are also aligned along the out-of-plane direction. The conductivity of T-ECH in an out-of-plane direction is 0.56 S/cm. The conductivity of T-ECH in an out-of-plane direction was measured with a thick film of T-ECH.

=> To further investigate the anisotropic mechanical properties, frequency sweep measurements with two different directions were conducted. As expected, G' and G'' of T-ECH measured in-plane direction was higher than that of out-of-plane direction due to the more formation of PEDOT:PSS fibers in the in-plane direction.

Fig. r2| Frequency sweep of T-ECH in in-plane and vertical directions.

Rheological properties were measured with an oscillatory rheometer (MCR302, Anton Paar) using 8 mm parallel plate-plate geometry. Frequency sweep measurements were conducted with 0.25 N force applied with 1% shear strain.

Please elaborate on the nearest-neighbor hopping across the PSS-PAA lamellas and how does that compare with pure electrically conductive hydrogel previously reported?

=> We thank the reviewer for this suggestion and comment.

=> The PAA template physically confines the PEDOT to grow and makes a thin fibrous 3D structure. Therefore, the hole conduction in T-ECH is most likely to occur along the PEDOT chains via intra-charge and inter-charge transports. PEDOT chains in T-ECH have linearly extended conformations while those in Pure ECH have coiled conformations. It can be expected that intra-charge transport of T-ECH is much faster than that of Pure ECH. In addition, PEDOT chains have close proximity to each other through strong π - π interaction. Therefore, PEDOT-PEDOT staking distance in T-ECH is remarkably reduced compared to Pure ECH as seen in WAXS analysis (Fig. S13). The increased interchain packing enables efficient inter-chain charge transfer (*Nat. Commun.* **7**, 11287 (2016), *Phys. Rev. Lett.* **109**, 106405 (2012)).

Consequently, T-ECH can have high electrical conductivity through facilitated intra- and inter-chain charge transfer through continuously connected linear PEDOT chains. The hole hopping across the PSS-PAA lamellas could occur when they are close to each other, but the aforementioned mechanisms (hole hopping through the continuously connected PEDOT network via intra-, inter-chain transport) would dominate the hole conduction in T-ECH.

=> Ion conduction in T-ECH is most likely to occur along the PSS and PAA chains since it would have the lowest hopping activation energy. The lamella of PSS and PAA has uniform 3D connected structure which makes fast ion transport along their network. In addition, the electrostatic interactions between hole and PSS anions can be further shielded by PAA, which increases ion mobility. Since T-ECHs have large water content, ions can also be transported by water-rich domain.

Supplementary Figure 9 | Electrically conductive paths made in T-ECH. a, Schematic illustrating of inter-, intra-charge transfer in the conductive polymer. **b**, Schematic illustrating of intra-transfer pathways made in linearly connected PEDOT networks in T-ECH along the template PAA network. Because PEDOT in T-ECH is unentangled and has a linear

conformation, the intra-charge transfer is facilitated. The unentangled PEDOT chains have a high degree of intrachain and interchain ordering, which facilitates electrical conduction. The conductive pathways are well-made to have high electrical and ionic conductivities.

Supplementary Figure 14| WAXS profile of fully dried Pure ECH and T-ECH. The peak around $q = 1.85 \text{ \AA}^{-1}$ arises from face-to-face stacking of PEDOT thiophenes. Therefore, the π - π stacking distance of PEDOT can be measured from the q value. The peak q values of Pure ECH and T-ECH are 1.87 and 1.91 \AA^{-1} , respectively. Therefore, π - π stacking distances of Pure ECH and T-ECH are 3.36 \AA and 3.28 \AA , respectively ($d = 2 * \pi/q$). The smaller distance between thiophenes in T-ECH indicates that PEDOT-PEDOT interaction is stronger in T-ECH than that of Pure ECH, which results in more stable electrical pathways in the hydrogel state. This phenomenon comes from the confined PEDOT nanofibers in PAA.

Main text, Page 11

The Pure ECH peak around $q = 1.85 \text{ \AA}^{-1}$ arose from π - π stacking of PEDOT thiophenes, whereas the π - π stacking peak of T-ECH occurred at a higher q (1.87 \AA^{-1} to 1.91 \AA^{-1}), indicating a closer PEDOT-PEDOT distance (3.36 \AA to 3.28 \AA). Thus, the PEDOT-PEDOT interactions were stronger and more well-ordered in T-ECH, which resulted in a more stable and

homogeneous linear PEDOT network without conformational disorders. Due to the increased interchain packing of PEDOT, it could have a high inter-chain charge transfer. Consequently, T-ECH can have high electrical conductivity through facilitated intra- and inter-chain charge transfer through continuously connected PEDOT chains.

=> Compared to the previously reported PEDOT:PSS hydrogel (Pure PEDOT:PSS hydrogels. *Nat. Commun.* **10**, 1043 (2019)), in T-ECH, the hole conduction is less disrupted by PSS-rich insulating bulk domains. Pure PEDOT:PSS hydrogel has large PSS aggregations that disrupt the long-range assembly of PEDOT chains. As shown in AFM images, there are clear phase separations of the PEDOT-rich domain and PSS-rich domain. Thus, the inter-chain charge transfer is limited due to the presence of PSS-rich domains. However, in the case of T-ECH, both PEDOT and PSS grow along the porous structure of PAA, leading to a homogeneous thin 3D fibrous network with fully extended PEDOT:PSS chains. Therefore, PEDOT: PSS fibers in T-ECH are well interconnected. In addition, pure PEDOT:PSS hydrogel would have more energetic disorders due to the entangled bulk PEDOT domains, which is unfavorable for charge transport (*Nat. Mater.* **12**, 1038–1044 (2013), *Adv. Funct. Mater.* **23**, 6024–6035 (2013)). Pure PEDOT:PSS hydrogel has more disordered structures in aggregation than T-ECHs. Fig. 3a is modified to give more information about this mechanism and explained in more detail.

Fig. 3a| Schematic illustration of microstructures of Pure ECH, T-ECH, and PAAm-

based T-ECH (PAAm T-ECH). In Pure ECH, the electrically conductive pathway is disrupted by insulating PSS-rich domains. On the other hand, T-ECH has a continuous PEDOT-connected network without bulk PSS aggregates limiting the conductive pathway. PAAm-based T-ECH cannot have PEDOT fibrous network due to the phase separation of PEDOT:PSS and PAAm.

Main text, Page 9-10

The major reason for its high electrical performance is the elimination of bulk PEDOT aggregates in the percolation network and the formation of a thin, extended fibrous structure of PEDOT chains (Fig. 3a). In Pure ECH, PEDOT and PSS bulk domains are inevitably formed. Thus, it has microscale PEDOT:PSS-rich aggregates, as confirmed by microcomputed tomography (Supplementary Fig. 8). Since Pure ECH has a large number of PSS aggregates that disconnect the PEDOT domains, the PSS-rich domains significantly block the conductive path. Therefore, the PEDOT-PEDOT interdomain hole transfer is limited. Also, highly entangled PEDOT networks with extensive conformational disorder work as bottlenecks of efficient charge transport in Pure ECH³⁰. Consequently, the conductivity cannot exceed 40 S/cm due to the limited transfer path. In sharp contrast, T-ECH shows a homogeneous PEDOT:PSS nanofibrous network without microscale phase separation inside the hydrogel (Supplementary Fig. 8b). During the synthesis of T-ECH, both PEDOT and PSS fibers grow only through the porous PAA template network. Therefore, PAA template physically confines the PEDOT:PSS fibers in T-ECH to enable a well-connected fibrous network. Accordingly, insulating PSS doesn't make large domains that block the conductive PEDOT pathway. Therefore, continuously connected PEDOT path enables high electrical conductivity. In addition, the fine-extended PEDOT:PSS fibers have fast and efficient charge transport via intra-chain charge transport (Supplementary Fig. 9 and 10)^{30,31,32}. The intra-chain charge

transport in Pure ECH is inefficient due to the entangled PEDOT chains. However, T-ECH has linear and extended chain conformations leading to fast intra-chain charge transfer.

4. Based on previous research works on PEDOT:PSS (e.g., J. Rivnay Nat. Commun.2016, 7, 11287.) the hydrogel is expected to have a PEDOT:PSS-rich fiber (towards the ultimate limit of 1:1 PEDOT:PSS), and PSS-PAA lamellas. Please consider to change the term “PEDOT fiber” to “PEDOT:PSS fiber” as PEDOT is doped by PSS, and modify the schematic drawing on Fig 3a.

=> We thank the reviewer for this suggestion and comment. We have changed the “PEDOT fiber” to “PEDOT:PSS fiber”. According to the reviewer’s comment, we modified Fig. 3a.

Fig. 3a| Schematic illustration of microstructures of Pure ECH, T-ECH, and PAAm-based T-ECH (PAAm T-ECH). In Pure ECH, the electrically conductive pathway is disrupted by insulating PSS-rich domains. On the other hand, T-ECH has a continuous PEDOT-connected network without bulk PSS aggregates limiting the conductive pathway. PAAm-based T-ECH cannot have PEDOT fibrous network due to the phase separation of PEDOT:PSS and PAAm.

5. The author explained that continuous breakage/reformation of hydrogen bonds between PEDOT:PSS network and PAA template contributes to the stretchability. Please characterize or explain the viscoelasticity or viscoplasticity of the materials. To what degree the deformation is irreversible? How does the stress-strain curve response to different strain rate.

=> We thank the reviewer for this suggestion and comment.

=> The reversible hydrogen bonding between PSS and the PAA template makes the network tougher. However, there is a small amount of PEDOT:PSS in the T-ECH, so it exhibits more elasticity than viscosity. From the rheological frequency sweep measurement, the elastic modulus (G') is shown to be higher than the loss modulus (G'') over a wide range of frequencies, meaning that the T-ECH exhibits elastic hydrogel-like mechanical behavior.

Fig. S10a| Frequency sweep measurements of T-ECHs. Rheological properties were measured with an oscillatory rheometer (MCR302, Anton Paar) using 8 mm parallel plate-plate geometry. Frequency sweep measurements were conducted with 0.25 N force applied with 1% shear strain.

=> We've also done a cyclic tensile test to investigate the elastic recovery of T-ECH (strain rate = 100%/min). When stretched to 50%, T-ECH showed the negligible residual strain. At

first stretching-releasing cycle, hysteresis may result from reversible breakage of PAA-PSS hydrogen bondings and PEDOT-PEDOT interaction. Also, when the tensile strain is 200%, the residual strain is less than 50%, meaning that it has elastic properties (*Nat. Commun.* **13**, 358 (2022)). Although T-ECH has a double-network structure of PEDOT:PSS and PAA, the composition of PEDOT:PSS is much lower than PAA. Accordingly, the mechanical destruction while stretching is low due to the elastic nature of the PAA template. Therefore, it can have low residual strain and can be used for practical applications.

Fig. S7| Cyclic tensile test of T-ECH. Cyclic tensile tests were conducted with a universal testing machine (Instron 68SC-1). Samples with dimensions of 5×10 mm (width \times length) were measured. **a**, Cyclic tensile test was done for 5 cycles with 50% strain and a strain rate of 100%/min. **b**, Cyclic tensile test with different strains (50%, 100%, and 200%) with a strain rate of 100%/min.

=> Tensile tests with different strain rates were conducted with T-ECH (T-ECH 2). When the strain rate increased from 50%/min to 200%/min, the stretchability of T-ECH slightly decreased. However, it still shows high stretchability over 400% strain. When the strain rate is high, there is not enough time to dissipate the strain energy by reorganization of its network,

leading to lower fracture strain. There was no significant change in the initial modulus of T-ECH with different strain rates.

Fig. r1| Tensile test on T-ECH with various strain rates.

Tensile tests were done with a universal testing machine (UTM) (Instron 68SC-1). Samples with dimensions of 5 × 10 mm (width × length) were measured with strain rates of 50%/min, 100%/min, and 200%/min.

=> Additionally, we saw that T-ECH restored its initial state after large compression, as seen in Fig. 2d and Supplementary Movie 1.

Fig. 2d| Images of compressed PAA and T-ECH with 60 kPa (20 N) and 120 kPa (35 N), respectively.

Supplementary Movie 1| Compressing T-ECH with a 1 MPa stress.

Minor Points:

1. How does dry-annealed at 95 °C improve the conductivity and mechanical properties? Can you explain the rationale for dry-annealing?

=> We thank the reviewer for this suggestion and comment. The purpose of DMSO addition is the destabilization of PEDOT:PSS colloids. Electrostatic interactions between PEDOT and PSS can be decreased by DMSO. At the same time, electrostatic repulsion between PEDOT:PSS colloids can be decreased as well. During the solvent evaporation and dry-annealing process, fibrous structures start to form through π - π interaction of PEDOT chains, and they are interconnected. Thus, PEDOT-PEDOT connected network can be made inside the template polymer.

=> High conductivity is derived from the well-defined fibrous network of PEDOT:PSS in the template. We found that PAA template enables the formation of well-defined percolation network with very thin and extended PEDOT fibers. Without a PAA template, only thick PEDOT:PSS aggregates with bulk structure are observed.

=> Without a well-defined fibrous network of PEDOT:PSS in PAA template, PAA hydrogel cannot be mechanically toughened. Energy dissipative PEDOT:PSS network is a key for the high toughness of hydrogel. Pre T-ECH, which has only PEDOT:PSS colloids, is not mechanically tough due to the lack of energy dissipative network.

Main text, Page 6-7

In this state (Pre T-ECH), PEDOT:PSS is in colloidal form with hydrophobic PEDOT in the core surrounded by an anionic PSS shell. Subsequently, DMSO was added to the Pre T-ECH to induce the transformation of PEDOT:PSS from a colloids to extended nanofibers by disturbing the ionic interaction between PEDOT and PSS. Once the linear PEDOT chains are made, they can be connected to each other through π - π interaction. To facilitate their connection, all the solvents are removed through dry-annealing to make PEDOT fibers encounter each other. Therefore, a uniformly connected PEDOT network is assembled along the PAA template chains. Finally, it was re-swelled in water to form a highly conductive and mechanically tough hydrogel (T-ECH).

2. “T-ECH can endure a high compressive force (35 N)”: I suggest the authors to use areal normalized parameter, i.e., stress instead of force to describe the mechanical properties.

=> We thank the reviewer for this suggestion and comment. We have changed the force to stress.

Main text, Page 8

In addition, similar to other double-network hydrogels, T-ECH can endure high compressive stress (Fig. 2d, e; Supplementary movie 1).

3. Referring to the sentence “Therefore, the fine extended thin PEDOT fibers have fast and efficient charge transport via intra-chain transport, which is faster than inter-chain transport (Supplementary Fig. 8 and 9)”, only supplementary Fig. 8 illustrate the inter-chain transport. Fig S9 is rheology comparison between T-ECHs and Pure ECH. The authors should add explanation about how lower degree of entanglements from Fig. S9 contributes to more intra-

chain transport.

=> We thank the reviewer for this suggestion and comment. From the rheological measurement (Fig. S10b), it is shown that the storage modulus of T-ECH is much lower than Pure ECH, implying that T-ECH has a less entangled network than Pure ECH, which has only highly entangled PEDOT:PSS aggregates. PEDOT:PSS fibers are thin and well-extended in T-ECH. In contrast, fibers in Pure ECH are very thick, meaning that many PEDOT chains are entangled inside. Therefore, electrical conduction in T-ECH is faster than in Pure ECH due to the high intra- and inter-chain ordering in PEDOT:PSS fibers. We added this information in Supplementary Fig. 10.

Fig. S10b| Frequency sweep measurements of T-ECH and Pure ECH. The storage modulus of T-ECH is lower than Pure ECH, implying that T-ECH has less entangled network than Pure ECH which has only highly entangled PEDOT:PSS aggregates. Therefore, electrical conduction in T-ECH is faster than in Pure ECH due to the high intra- and inter-chain ordering.

4. There is a typo in Fig S10 caption “Therefore, we can conclude that PEDOT in T-ECH is in linear and thin fibrous network, network,” There are two “network”s

=> We thank the reviewer for the correction. We have changed it.

Supplementary Information, Page 13-14

Therefore, we can conclude that PEDOT in T-ECH has a linear and thin fibrous network, contrary to Pure ECH.

5. There is a typo in the following sentence “Compared with PEDOT:PSS, T-ECH had a drastically increased ration of PEDOT to PSS from 0.53 to 0.88”. Ration > Ratio

=> We thank the reviewer for the correction. We have changed it.

Main Text, Page 12

Compared with PEDOT:PSS, T-ECH had a drastically increased ratio of PEDOT to PSS from 0.53 to 0.88, implying that an excess amount of insulating PSS shell was washed out during the template-directed assembly of the highly conductive PEDOT:PSS fibrous network³⁸.

Reviewer #3

Chong et al here report on a type of electrically conducting hydrogel comprised of Polyacrylic Acid and nanostructured PEDOT:PSS. With their method, a highly stretchable and electrically conductive hydrogel is achieved, that finds potential use in bioelectronic applications. The work is based on their previously reported nano-confinement method for obtaining a well-ordered PEDOT network. This has now been demonstrated to also work within a PAA hydrogel. With this, conductivities on the order of 240 S/cm were achieved, which are very high for polymer-based hydrogel composites.

However, several issues exist with the current version of the manuscript.

=> We have now carefully revised the manuscript according to the reviewer's comments and suggestions. All main revisions are marked in red font.

For once, the title prominently suggests the existence of a metallic nano fibrous network, where in reality a conductive polymer based network is used. This is reasoned later on in the text with the measured conductivity, and the relatively low mass percentage of PEDOT:PSS that is hypothesized to imply metallic conductivity in said PEDOT. No direct proof of that hypothesis is given. Then, quite misleadingly, an ashby plot (Figure 3 g) is labeled "Ashby Plot of conducting hydrogels", where in reality only polymer-conductor-based hydrogels are shown in that plot (an information stated in the caption, but that is really misleading). Consequently, stretchable conducting hydrogels based i.e. on (actually metallic) silver nanostructures are omitted from the plot. Such networks exhibit even higher conductivity (>350 S/cm, in comparison to the here reported 247 S/cm).

=> We thank the reviewer for this suggestion and comment. In T-ECH, 99% of its composition is a non-conductive part (water and PAA). Assuming that PEDOT:PSS accounts for 1% of the

total volume and the electrical conduction happens only through the PEDOT, it can be said that PEDOT:PSS in T-ECH shows extremely high electrical conductivity. However, the charge transport mechanism in our hydrogel follows the same as other conductive polymers. Therefore, we changed the term “metallic” to “highly conductive”. Since our material is a hydrogel (90% water), we couldn’t perform temperature-dependent resistivity measurements (evaporation problem). Our PEDOT:PSS fibrous networks show just efficient charge transport due to well-defined microstructure and have relatively high charge carrier concentrations (high doping level). As a result, we achieved the highest electrical conductivity among conductive polymer-based hydrogels. Some dried PEDOT:PSS polymer thin films (0% water content) show higher electrical conductivity (>4,000 S/cm).

=> As in the introduction part, conductive polymer hydrogels have excellent biocompatibility to use in bioelectronic applications. They are experimentally confirmed by many research groups. However, in the case of conductive hydrogels that use metal additives such as silver flakes, the metal particles can be oxidized and release metal ions that induce cytotoxicity. Therefore, such cytotoxic metals cannot be used for the tissue-interfacing electrode (*ACS Nano* **3**, 279–290 (2009), *Acta Biomaterialia* **10**, 439-449 (2014), *Journal of Advanced Research* **9**, 1-16 (2018)). Thus, we considered only conductive polymer hydrogels since this work focuses on the bio-interfacing electronic materials. Also, for bioelectronic applications, not only electrical conductivity but also ionic conductivity is important because the bio-system is going through ionic signal transmissions. The conductive polymer hydrogel has superior ionic conductivity and high volumetric capacitance due to the molecular level electrical double layer (EDL) formation . Therefore, we considered only conductive polymer-based hydrogels in Fig. 3g, excluding the ones that used metal fillers since we are synthesizing material for bio-interfacing devices. We have added this information in the introduction part for a more precise understanding. However, we understood this plot causes misunderstanding about conductive

hydrogels. Thank you for your comment. We changed Fig. 3g and added some metal-based conductive hydrogels.

Fig. 3g| Ashby plot of conductive hydrogels. PEDOT:PSS-based hydrogel (blue), polyaniline (PANI)-based hydrogel (yellow), polypyrrole (PPy)-based hydrogel (green), and metal-based hydrogel (gray) (Supplementary Fig. 1).

Main text, Page 3

However, conductive additives such as metal, CNT, or graphene cannot be employed since they cause adverse reactions in the body due to their cytotoxicity. Therefore, conductive polymer hydrogel is considered a promising material for tissue-interfacing electrodes due to its biocompatibility, tissue-like mechanical properties, mixed electron/ion conduction, and water-rich nature^{1,6,7}.

This is even more surprising, as the corresponding work is then found as reference in the supplemental material (1. Ohm, Y. et al. An electrically conductive silver–polyacrylamide–alginate hydrogel composite for soft electronics. *Nat. Electron.* 4, 185–192 (2021)). Now,

talking about metallic nano fibrous networks in the title, but then leaving out actual metal-based conducting hydrogels in the comparison is highly dubious. If such comparison plots are included, they should be balanced and fair. Metallic conductivity in the polymer network should be unambiguously proven if claimed. In the methods, the authors mention the use of 4-point probes (M4P-205 System, MS TECH) that press onto the hydrogel (with probe heads very close to each other) to measure conductivity. Specifics on that setup would be helpful. They are typically used to measure sheet resistance in thin solid films. Is the hydrogel compressed during measurement? conductivities in excess of 2400 S/m in PEDOT:PSS thin films have been reported. Such a setup seems not very suitable to measure conductivity in large, soft sheets of hydrogels, there may be a source of significant error here.

=> We thank the reviewer for this suggestion and comment.

Fig. S2b| A comprehensive comparison between T-ECH of this work and previously reported conductive hydrogels in terms of electrical conductivity, ionic conductivity, softness, toughness, adhesiveness, and biocompatibility including pure PEDOT:PSS hydrogel¹⁸, soft hydrogel², Ag-hydrogel¹, and adhesive¹⁹.

As mentioned, the metal additives such as silver flakes can be oxidized and release metal ions that induce cytotoxicity and thus cannot be used for the tissue-interfacing in-vivo electrode. In Fig. S2b, “Ohm, Y. et al. An electrically conductive silver–polyacrylamide–alginate hydrogel

composite for soft electronics. *Nat. Electron.* **4**, 185–192 (2021)” was added as a reference for comparison that this material lacks biocompatibility, adhesiveness, and stability in physiological conditions. Ag-hydrogel has a great advantage in terms of electrical conductivity and softness, it may be suitable for wearable and soft robotic applications rather than implantable applications.

Nat Electron 4, 185–192 (2021), Fig. S6

Also, in “*Nat. Electron.* **4**, 185-192 (2021)”, it is said that silver–polyacrylamide–alginate hydrogel composite is made by partial dehydration. This implies that in the wet environment, it would absorb water significantly, which is unstable for electrical and mechanical properties as in Fig. S6 in the “*Nat. Electron.* **4**, 185-192 (2021)” paper. Therefore, its conductivity may continuously increase in vivo.

=> For the conductivity measurement with the 4-point probe, we added more specific information in the method section. PEDOT:PSS hydrogels were cut into a rectangular shape (10 mm in length, and width of 5 mm in width). It was measured with the stainless-steel wire electrodes attached to the surface of the hydrogel without pressing and connected to the probe tips (*Nat. Commun.* **10**, 1043 (2019)).

Fig. r1| Electrical conductivity measurement setup.

Main text, Page 25-26

Electrical characterization

Electrical conductivity was measured with a 4-point probe (M4P-205 System, MS TECH) connected to a source meter (2400 SourceMeter, Keithley). The conductivity was measured after swelling in D.I. water or PBS for 24h. The measured T-ECH had a length of 10 mm and a width of 5 cm. The thickness of T-ECH 1 was 600 μm, T-ECH 2 was 800 μm, and T-ECH 3 was 1 mm. It was measured with the stainless-steel wire electrodes attached to the surface of the hydrogel and the wires were connected to the probe tips.

Related, the key figures of merit (conductivity, toughness) lack error bars, the statistics is unclear (Fig 2f).

=> We thank the reviewer for this suggestion and comment. Conductivity and mechanical measurements were performed with 5 samples and 3 samples, respectively. We would add this information in Fig. 2f and add error bars.

Fig. 2f] Graph comparing electrical conductivities and toughness of Pre T-ECH, T-ECH, and Pure ECH. The data plotted represents the mean and standard deviation (n = 5 for conductivity, n = 3 for toughness).

Then, the term "Adhesive-free" in the title implies that no further treatment of the hydrogel is needed in order for it to adhere to various surfaces. This is contradicted by the need of an "activation layer" (Figure 4a, methods: "An aqueous mixture of EDC (15 mg/mL) and NHS (23 mg/mL) was dropped on T-ECH to make T-ECA", gold surfaces had to be functionalized as well to allow for adhesion). Now, it is perfectly fine to do such modifications to achieve tough adhesion. But "adhesive-free" should then not be used, because it is again misleading.

=> We thank the reviewer for this suggestion and comment. We used the term "adhesive-free" since activation took place inside of T-ECH without any additional layer formation. Our T-ECH is intrinsically adhesive to tissue surface by H-bonding and electrostatic interactions. In

order to introduce covalent crosslinking between hydrogel and tissue for long-term applications, we activated the surface of T-ECH without making an additional layer. At a molecular level, the Carboxylic acid groups of PAA were activated by NHS-ester. However, to avoid misunderstanding, we would delete “adhesive-free” in the title. The title is changed to “**Highly conductive tissue-like hydrogel interface through template-directed assembly**”.

The mechanical data on the cyclic fatigue tests should be shown as well, not only the electrical one. I wonder if the nano-structured PEDOT network fatigues. Corresponding SEM images (as done in figure S14) after mechanical fatigue might also be helpful here.

=> We thank the reviewer for this suggestion and comment. We’ve done a cyclic tensile test to investigate the elastic recovery of T-ECH (strain rate = 100%/min). When stretched to 50%, T-ECH showed a negligible residual strain. Also, even when the tensile strain is 200%, T-ECH showed good elastic recovery (*Nat. Commun.* **13**, 358 (2022)). At the first stretching and releasing cycle, we observed relatively large hysteresis. It may be due to the breakage of PEDOT:PSS network and PSS-PAA hydrogen bonds. However, we still observed stable electrical properties of our T-ECH up to 300% strain.

Fig. S7| Cyclic tensile test of T-ECH. Cyclic tensile tests were conducted with a universal testing machine (Instron 68SC-1). Samples with dimensions of 5 × 10 mm (width × length)

were measured. **a**, Cyclic tensile test was done for 5 cycles with 50% strain and a strain rate of 100%/min. **b**, Cyclic tensile test with different strains (50%, 100%, and 200%) with a strain rate of 100%/min.

Fig. 2g| Resistance change of T-ECH 1 under strain. Strain rate = 200%/min.

=> As the reviewer pointed out, PEDOT:PSS network may be destructed by stretching-releasing cycles (fatigue). However, the electrical resistance of PEDOT:PSS network can be maintained for more than 100 stretching-releasing cycles at 100% strain.

Supplementary Fig. 6d| Resistance change to the cyclic strain of 100%. Strain rate = 300%/min.

=> We've also done SEM on the stretched T-ECH (100% strain). No significant phase segregation or structural destruction was observed from the SEM image. Due to the resolution limitation of SEM, PEDOT:PSS and PAA domains cannot be seen separately. It shows stretched polymer networks along the direction of the strain when stretched 100%. After stretching and releasing cycle, T-ECH has been shown to reform its network. Samples are prepared by freeze-drying process of T-ECHs.

Fig. r3| Cross-sectional SEM images of freeze-dried (a) T-ECH, (b) stretched T-ECH, and (c) stretched and released T-ECH.

Hydrogels were freeze-dried by freezing into liquid nitrogen and drying in freeze-drier under vacuum conditions of 4×10^{-3} Torr. Cross-sectional images were obtained using SEM (S-4800,

Hitachi).

Figure S6 d-f, what are the values for the initial resistance in those cases? (in Ohms, not the normalized values). What are the dimensions of the samples? Again, more information on how conductivity was actually measured would be very valuable.

=> We thank the reviewer for this suggestion and comment. The initial resistance of T-ECH 1 was 5 Ω , 2 was 108 Ω , and 3 was 82 Ω . The dimensions are 10 mm in length, and 5 mm in width. The samples were placed on the stretcher and were connected to the source meter using stainless-steel wires. We added this information in the description of Fig. S6 and the method section.

SI Page 8

Fig. S6| Resistance change during stretching and cyclic stretching. a, b, c, Resistance change during the strain of T-ECHs. Strain rate = 200%/min. d, e, f, Resistance change during the cyclic strain of 100%. Strain rate = 300%/min. ($R_0 = 5 \Omega$ (T-ECH 1), 108 Ω (T-ECH 2), 82 Ω (T-ECH 3).

Main text, Page 25-26

Electrical characterization

Electrical conductivity was measured with a 4-point probe (M4P-205 System, MS TECH) connected to a source meter (2400 SourceMeter, Keithley). The conductivity was measured after swelling in D.I. water or PBS for 24h. The measured T-ECH had a length of 10 mm and a width of 5 cm. The thickness of T-ECH 1 was 600 μm , T-ECH 2 was 800 μm , and T-ECH 3 was 1 mm. It was measured with the stainless-steel wire electrodes attached to the surface of

the hydrogel and the wires were connected to the probe tips.

In conclusion, this work needs at least mayor revisions. The measured conductivity values have to be unambiguously proven, and correctly benchmarked to conducting hydrogels at large.

=> We thank the reviewer for this suggestion and comment. The conductivity was measured using the same method employed in other conductive hydrogel papers (*Nat. Commun.* **10**, 1043 (2019), *Sci. Adv.* **6**, eaay5394 (2020), *Sci. Adv.* **3**, 1602076 (2017), *Adv. Mater.* **34**, 2200261 (2022)). The in-vivo ultra-low voltage neurostimulation and high SNR of ECG recording with excellent biocompatibility are achieved due to the high electrical conductivity and tissue-like mechanical properties of T-ECH.

Reviewers' Comments:

Reviewer #1:

Remarks to the Author:

All comments from the reviewer were addressed properly in the revised manuscript which is now ready for publication.

Reviewer #2:

Remarks to the Author:

In the rebuttal letter, Chong et al. provided comprehensive responses to all the questions and concerns I raised. They substantiated their claims with additional data and explanations, which have greatly strengthened the manuscript. Based on their responses and the revised version of the manuscript, I recommend the publication of this work.

Reviewer #3:

Remarks to the Author:

The authors have extensively revised their manuscript. Several misleading statements have been removed, and measurements have been clarified. Overall, the study is now much more solid, and will be a nice contribution to polymer based conducting hydrogels.

Point-to-Point **Responses** to Reviewers' Comments

Reviewer #1 (Remarks to the Author):

All comments from the reviewer were addressed properly in the revised manuscript which is now ready for publication.

Reviewer #2 (Remarks to the Author):

In the rebuttal letter, Chong et al. provided comprehensive responses to all the questions and concerns I raised. They substantiated their claims with additional data and explanations, which have greatly strengthened the manuscript. Based on their responses and the revised version of the manuscript, I recommend the publication of this work.

Reviewer #3 (Remarks to the Author):

The authors have extensively revised their manuscript. Several misleading statements have been removed, and measurements have been clarified. Overall, the study is now much more solid, and will be a nice contribution to polymer based conducting hydrogels.

=> We truly thank all the reviewers for carefully reviewing our manuscript and their valuable comments.